# A biobank-scale test of marginal epistasis reveals genome-wide signals of polygenic interaction effects

Boyang Fu [1,11], Ali Pazokitoroudi [2,11], Zhuozheng Shi [3,4], Asha Kar [5,6], Albert Xue[5], Aakarsh Anand [2], Prateek Anand [2], Zhengtong Liu[2], Richard Border [7], Päivi Pajukanta [5,6,8], Noah Zaitlen[6,9,10] & Sriram Sankararaman [2,6,10] ✉

The contribution of genetic interactions (epistasis) to human complex trait variation remains poorly understood due, in part, to the statistical and computational challenges involved in testing for interaction effects. Here we introduce FAME (FAst Marginal Epistasis test), a method that can test for marginal epistasis of a single-nucleotide polymorphism (SNP) on a quantitative trait (whether the effect of an SNP on the trait is modulated by genetic background). FAME is computationally efficient, enabling tests of marginal epistasis on biobank-scale data. Applying FAME to genome-wide association study (GWAS)-significant trait-SNP associations across 53 quantitative traits and ≈300 000 unrelated White British individuals in the UK Biobank (UKBB), we identified 16 significant marginal epistasis signals across 12 traits ($P < \frac{5 \times 10^{-8}}{53}$). Leveraging the scalability of FAME, we further localized marginal epistasis signals across chromosomes and estimated the proportion of variance explained by marginal epistasis effects. Our study provides evidence for interactions between individual genetic variants and polygenic background influencing complex traits.

Understanding the role of epistasis is important for elucidating the genetic architecture of complex traits[1]. Epistasis could potentially explain the missing heritability underlying complex traits[2] (although some studies suggest a limited contribution from genetic interactions[3,4]), why genetic effects differ across ancestral populations[5] and the lack of transferability of genetic predictors both within[6] and across ancestries[7,8]. Epistasis has also been hypothesized to play a role in variable expressivity of complex traits[9].

Nevertheless, our understanding of the role of epistasis in human traits remains limited[10].

One class of methods to detect epistasis explicitly searches for pairs of single-nucleotide polymorphisms (SNPs) that have a non-linear effect on a trait. Although allowing for an unbiased search for epistasis, exhaustively testing all pairs is computationally difficult and requires a stringent significance threshold to control the false positive rate. Efforts to overcome these difficulties have involved

[1]Department of Biomedical Informatics at HMS, Harvard University, Boston, MA, USA. [2]Department of Computer Science, University of California, Los Angeles, Los Angeles, CA, USA. [3]Graduate Group in Genomics and Computational Biology, Perelman School of Medicine, University of Pennsylvania, Philadelphia, PA, USA. [4]Department of Genetics, Perelman School of Medicine, University of Pennsylvania, Philadelphia, PA, USA. [5]Bioinformatics Interdepartmental Program, University of California, Los Angeles, Los Angeles, CA, USA. [6]Department of Human Genetics, David Geffen School of Medicine at UCLA, Los Angeles, CA, USA. [7]Department of Computational Biology, CMU, Pittsburgh, PA, USA. [8]Institute for Precision Health, David Geffen School of Medicine at UCLA, Los Angeles, CA, USA. [9]Department of Neurology, University of California, Los Angeles, Los Angeles, CA, USA. [10]Department of Computational Medicine, David Geffen School of Medicine at UCLA, Los Angeles, CA, USA. [11]These authors contributed equally: Boyang Fu, Ali Pazokitoroudi. ✉e-mail: sriram@cs.ucla.edu

the use of statistical[11-13] and algorithmic techniques[14,15], hardware infrastructure[16-18] or a biologically informed reduction in the space of SNP pairs tested[19-23]. An alternate approach to detect epistasis aims to test for the aggregate epistatic effect across SNPs, often using a variance component framework[24-26]. In this framework, models of marginal epistasis[25,27] aim to test if the effect of an SNP on a trait is modulated by an individual's polygenic background. Such tests can improve power on account of the reduced multiple testing burden and the aggregation of weak epistatic signals. Even with the potential improvements in power, detecting robust signals of marginal epistasis likely requires application to large sample datasets[1,4] that are now becoming available[28,29]. However, estimating marginal epistasis from biobank-scale data sets is computationally challenging.

To address this challenge, we propose FAst Marginal Epistasis test (FAME), an algorithm that can jointly estimate additive and marginal epistasis variance components that is efficient. The efficiency of FAME enables its application to estimate marginal epistasis at a target SNP paired with genome-wide SNPs across a large number of individuals. We performed extensive simulations to show that FAME provides calibrated tests of marginal epistasis, is robust to model misspecifications and has adequate power to detect true marginal epistasis signals. We applied FAME to test for marginal epistasis at trait-associated genome-wide association study (GWAS) SNPs for 53 quantitative traits in the UK Biobank (UKBB) ($N \approx 300\,000$ unrelated White British individuals and $M \approx 500\,000$ SNPs on the UKBB genotyping array) and attempted to replicate the discovered signals in the All of Us (AoU) dataset. To better characterize our findings, we attempted to partition marginal epistasis signals across chromosomes, estimate the proportion of variance explained by the marginal epistasis effects and interpret these signals using functional genomic data.

## Results

### Methods overview

We aim to test whether the effect of a target SNP on a phenotype is modulated by the genetic background of the individual by assessing whether the pairwise interactions of the target SNP with each of the remaining SNPs contribute, in aggregate, to variance in the phenotype. In contrast to testing for interactions at a chosen pair of SNPs, testing for marginal epistasis could potentially be more powerful when interaction effects are polygenic; that is, we have a large number of interactions each with a weak effect while also benefiting from the reduced multiple testing burden. To ensure that additive genetic effects are not incorrectly attributed to interactions, we jointly model the additive effects from all genome-wide SNPs (including the target SNP) in addition to the marginal epistasis effects.

FAME uses a variance components model in which phenotypic variance is partitioned into genome-wide additive genetic variance ($\sigma_g^2$), marginal epistasis variance at a target SNP $t$ ($\sigma_{gxg,t}^2$) and residual variance (see Fig. 1 for an example and Methods for details). The marginal epistasis variance component $\sigma_{gxg,t}^2$ captures the aggregate contribution of all pairwise interactions between the target SNP $t$ and the remaining SNPs in the genome. We would like to test whether the marginal epistasis variance component is significantly different from zero and to be able to estimate its value.

Fitting such a model to biobank-scale data, containing hundreds of thousands of individuals and millions of SNPs, is computationally challenging. Approaches that aim to efficiently compute genome-wide epistatic genomic relationship matrices[4] do not scale well with sample size while not directly estimating marginal epistasis. The CORE GREML approach[30] explicitly estimates covariance between random effects but does not directly model epistasis. MAPIT[25], a previous test for marginal epistasis, does not scale to large samples.

FAME expands on our recent work[31,32] to test and estimate marginal epistasis on biobank-scale data. Specifically, FAME utilizes a randomized Method-of-Moments (MoM) estimator that reduces the size

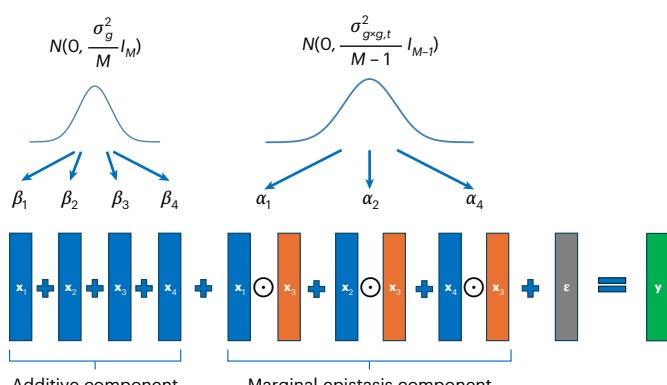

**Fig. 1 | The model underlying FAME.** In this example, we have genotypes at four SNPs denoted by $\mathbf{x}_1, \mathbf{x}_2, \mathbf{x}_3, \mathbf{x}_4$. We would like to test for marginal epistasis between SNP 3 (the target SNP) and the remaining SNPs. We model the relationship of the phenotype $\mathbf{y}$ to the genotypes as arising due to the additive effect of genotypes at each of the four SNPs, the pairwise interaction effects between genotypes at the target SNP ($\mathbf{x}_3$) and the remaining SNPs, and environmental noise $\boldsymbol{\varepsilon}$. The additive effect sizes $\boldsymbol{\beta}$ are drawn from a distribution with variance parameter proportional to $\sigma_g^2$, whereas the marginal epistasis effect sizes $\boldsymbol{\alpha}$ are drawn from a distribution with variance parameter proportional to $\sigma_{gxg,t}^2$ where $t = 3$ and $M = 4$.

of the input genotype and interaction matrices by multiplying each of these matrices with a prespecified number ($B$) of random vectors (each entry of the random vector being drawn independently from a standard normal distribution). This approach is used to approximate key computations in the MoM estimator (Methods). The accuracy of this estimator depends on $B$. We show that modest values of $B \approx 100$ lead to a method that yields accurate estimates while being highly scalable.

### Calibration of FAME

We assessed the false positive rate of FAME on simulated phenotypes with additive genetic effects but no genetic interactions. We simulated phenotypes based on genotypes from unrelated White British individuals in UKBB ($N = 291{,}273$ individuals, $M = 459{,}792$ SNPs). We set the proportion of trait variation explained by additive genetic effects (additive heritability) $\sigma_g^2 = 0.25$ and varied the proportion of variants that have non-zero additive effects (causal variants) $p \in \{0.01, 0.10\}$.

The key parameter in FAME is the number of random vectors $B$, which determines its scalability and stability ('Efficient computation of variance components' in Methods). We use $B = 100$ in all analyses while exploring the impact of this choice in 'Robustness of marginal epistasis signals'. We assessed the calibration of FAME when applied to two sets of target SNPs: target SNPs chosen randomly from the UKBB array and SNPs identified based on a GWAS to mirror our analyses of traits in UKBB ('Application to UK Biobank phenotypes').

Although FAME is calibrated when the target SNPs were selected at random (Supplementary Fig. 1), its $P$ values tend to be inflated when target SNPs were selected from a GWAS (Supplementary Fig. 2). To address this issue, we regressed out additive effects at SNPs that lie within the linkage disequilibrium (LD) block (identified in the 1000 Genomes European population[33]) around the target SNP and excluded these SNPs when constructing the set of genetic interactions (the $E_t$ matrix in Methods). This approach led to calibrated results (Fig. 2a).

### Robustness to model misspecification

To explore the impact of model misspecification on FAME, we considered scenarios in which (1) the relationship between covariates and phenotype is nonlinear, (2) the genetic architecture includes gene-environment interactions with a hidden environmental variable, and (3) the environmental noise is heavy tailed and (4) heteroskedastic

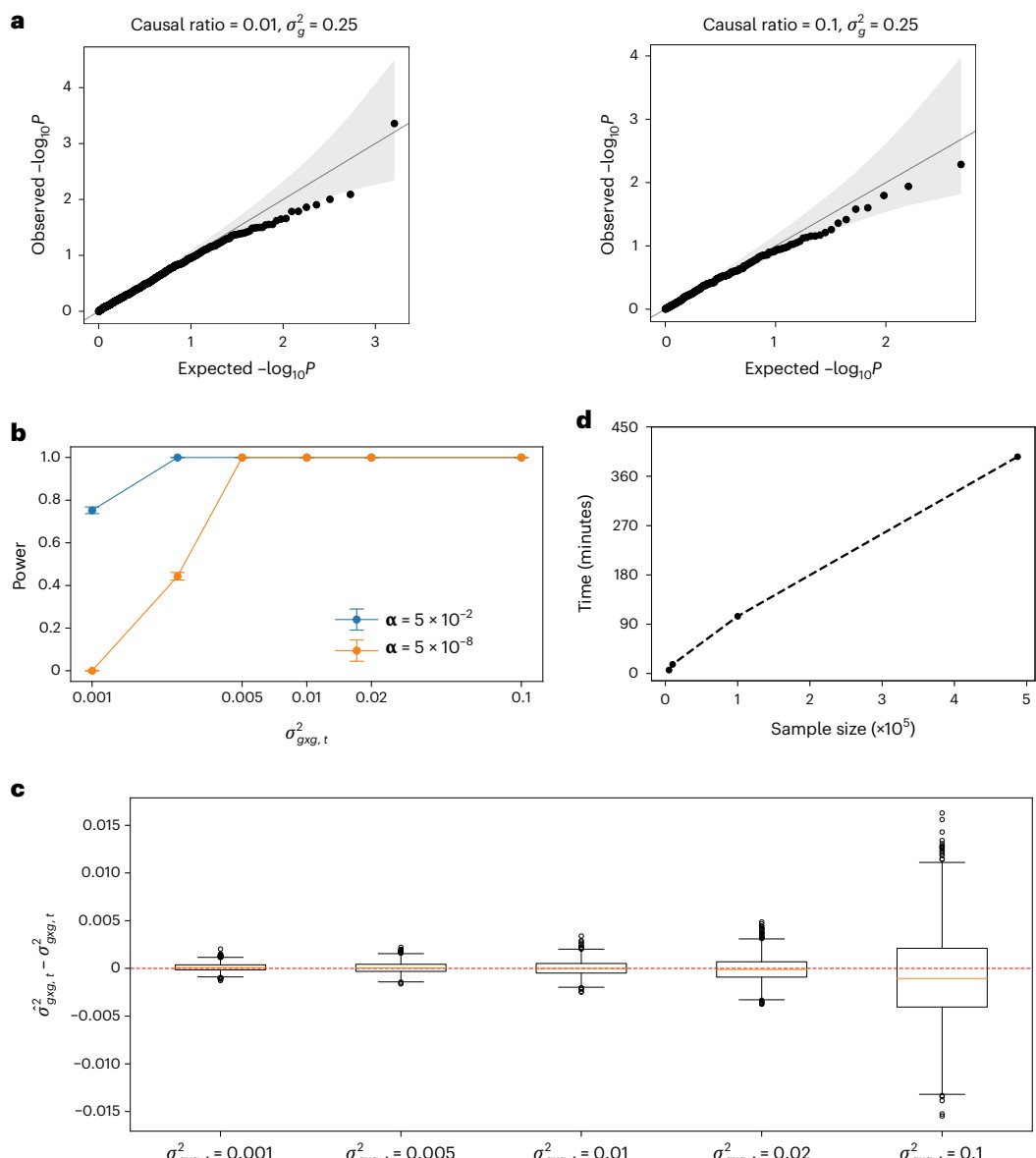

**Fig. 2 | Accuracy and runtime analysis of FAME. a**, Calibration of FAME. We applied FAME to simulated phenotypes with additive but no marginal epistasis effects. Phenotypes were simulated using genotypes measured on $N = 291,273$ unrelated White British individuals in the UK Biobank, with varying ratios of causal SNPs (causal ratio) and additive genetic variance component $\sigma_g^2 = 0.25$. We first ran GWAS to identify significant SNPs, which were then used as target SNPs in a marginal epistasis test. We find no significant marginal epistasis signals ($P < 5 \times 10^{-8}$) across any of the settings and that the $P$ values are calibrated with genomic inflation factor $\lambda_{gc} = 0.958\ (0.801, 1.06)$ and $1.053\ (0.754, 1.427)$ for causal ratio = 0.01 and 0.1, respectively. Shaded region shows the pointwise 95% confidence interval under the null, with bounds at the 2.5% and 97.5% quantiles of the Beta distribution. **b**, Power analysis of FAME. We simulated phenotypes by fixing the additive genetic variance component to 0.3 (approximately the median value estimated on real traits) across $N = 291,273$ unrelated White British

individuals in the UKBB-small dataset. We varied the magnitude of the marginal epistasis variance component ($\sigma_{gxg,t}^2$). We generated 1,000 replicates for each setting. We plot power for detecting marginal epistasis at a $P$ value threshold of 0.05 as well as the genome-wide threshold of $5 \times 10^{-8}$, with error bars representing the 95% bootstrap CIs from 10 K replicates. **c**, Accuracy of estimates of the marginal epistasis variance component ($\sigma_{gxg,t}^2$) in simulations. We used the same simulation as in **b**. We plot the error in the parameter estimates for each parameter setting. Box plots indicate the median (center line), the interquartile range (25th to 75th percentiles; box), and whiskers extend to the most extreme values within 1.5× the IQR from the quartiles across the 1,000 replicates. **d**, Runtime analysis of FAME. We computed the runtime of FAME applied to common SNPs on the UKBB whole-genome array data and varying sample size. We report the average across three runs at each setting.

(Supplementary Note 1). Prior work has shown that tests of epistasis can suffer from inflated false positive rates due to imperfect tagging of causal variants[34–36]. This led us to consider scenarios in which genetic variants have large additive effects and, additionally, not all such variants are observed. We simulated this scenario by drawing causal variants from imputed SNPs while we applied FAME to analyze SNPs on the UKBB array (section 'Datasets' in Methods and Supplementary Note 1).

Finally, we considered a scenario where the dependence of additive effect sizes on minor allele frequency (MAF) and LD has a different architecture[37] than the one assumed by FAME (Supplementary Note 1). We found that FAME remains calibrated across each of these settings (Supplementary Figs. 3, 4, 5 and 6, and Supplementary Tables 1 and 2).

Because it is of interest to compare the strength of marginal epistasis to marginal GWAS, we also evaluated the calibration of tests of the

ratio $\frac{\sigma^2_{gxg,t}}{\sigma^2_{gwas,t}}$ both when the additive model is correctly specified (Supplementary Note 2, Supplementary Fig. 7 and Supplementary Table 3) and under each type of model misspecification considered above (Supplementary Fig. 7 and Supplementary Table 3) to find that FAME produces calibrated $P$ values.

## Power analysis

To assess power, we simulated phenotypes with non-zero marginal epistasis variance components with $N = 291,273$ (section 'Simulations' of Methods). We observed that FAME has power ≥90% at $P < 5 \times 10^{-8}$ even when the variance explained by marginal epistasis is fairly low ($\sigma^2_{gxg,t} = 0.005$) (Fig. 2b). Further, FAME produced marginal epistasis estimates that were relatively unbiased (Fig. 2c).

## Computational efficiency

FAME can test marginal epistasis on 500 K individuals on a genome-wide dataset containing ≈500 000 SNPs in about 6 h with 32 GB RAM (Fig. 2d). We attempted to benchmark a previously proposed method, MAPIT[25] (section 'Runtime comparisons' of Methods) to find that its runtime grows rapidly even for modest numbers of individuals and SNPs: requiring more than 3 days to run on $N = 20$ K individuals with $M = 10\,000$ SNPs (Supplementary Fig. 8).

## Application to UK Biobank phenotypes

We applied FAME to 53 quantitative traits across $N = 291,273$ unrelated White British individuals (chosen to have adequate power and minimize population stratification) genotyped at common SNPs on the UKBB array. Our target SNPs were a set of 15,601 LD-pruned SNPs that were found to be associated with each trait in a GWAS (section 'UKBB GWAS' in Methods). We tested for marginal epistasis at the GWAS significant SNPs in which we also accounted for the additive effect of genome-wide SNPs and included age, sex and the top 20 genetic principal components (PCs) as fixed effect covariates. Our tests yielded 21 significant trait-SNP pairs across 13 traits ($P < \frac{5 \times 10^{-8}}{53}$ to account for the multiple traits tested; Fig. 3a). To additionally ensure that the additive genetic effects surrounding the target SNP do not impact estimates of marginal epistasis, we applied FAME to each of these 21 trait-SNP pairs after regressing out the additive effects of all of the SNPs in the LD block of the target SNP to observe 16 trait-SNP pairs across 12 traits that retain significant $P$ values for marginal epistasis ($P < \frac{5 \times 10^{-8}}{53}$; Table 1, additional information, including the marginal effect size of the target SNP, its nearest gene and the corresponding variant type in Table 2).

## Robustness of marginal epistasis signals

To explore the impact of the randomization underlying FAME, we selected two traits: body mass index (BMI) and serum urate levels (urate) as exemplary traits where FAME did not and did detect an epistatic signal. We experimented with the number of random vectors ($B$) and found that $B = 100$ yields consistent results (Pearson correlation of the negative log $P$ values across seeds $\rho = 0.99$ and = 0.98 for urate and BMI, respectively; Supplementary Fig. 9). We also found that the estimates of significant marginal epistasis signals were concordant across the number of PCs included (varying the number of PCs included from 20 to 40; Supplementary Fig. 10) and across the analysis of imputed SNPs compared to array SNPs (Supplementary Fig. 10; see Supplementary Note 3 for details). Additionally, we applied FAME to a permuted version of the epistatic matrix relative to the target variant to find that none of the SNP-trait pairs are significant (Supplementary Note 3 and Supplementary Table 4). Although phenotype scale can affect our results, traits are inverse rank normalized in all our analyses so that the results of FAME are invariant to monotone transformations (such as the scaling transformation considered here) and the $P$ values remain calibrated, as we confirm in simulations (Supplementary Note 3 and Supplementary Fig. 11).

## Validation and interpretation of marginal epistasis signals

We performed a series of analyses to validate our discoveries and attempted to interpret these signals.

## Replication of marginal epistasis signals

We assessed the robustness of the marginal epistasis signals detected by FAME using an internal and an external replication study. First, we split the UKBB unrelated White British individuals into a discovery and replication cohort of equal size. A GWAS followed by application of FAME to GWAS significant SNPs in the discovery cohort yielded three significant marginal epistasis trait-SNP pairs, all of which replicated ($P < \frac{0.05}{3}$; Fig. 4a). To validate our marginal epistasis signals in the AoU dataset[29], we identified five traits (of the 12 with at least one significant marginal epistasis SNP in UKBB) for which we had sufficiently large sample size of European ancestry ($N > 50\,000$) individuals in AoU. Applying a similar analysis as in UKBB (section 'Datasets' of Methods), we found that all five of our marginal epistasis signals replicated in AoU ($P < \frac{0.05}{5}$; Fig. 4b and Supplementary Table 5).

## Localizing signals of marginal epistasis

Having demonstrated evidence for genome-wide marginal epistasis, we sought to understand where these interactions localize. We extended FAME to test for marginal epistasis of a target SNP with only a subset of SNPs while accounting for the additive effects of genome-wide SNPs. For each target SNP, we tested, in turn, for marginal epistasis paired with other SNPs that fall on the same chromosome (local marginal epistasis; $gxg_{local}$) and with SNPs on chromosomes distinct from the chromosome containing the target SNP (distal marginal epistasis; $gxg_{dist}$). We confirmed that tests of $\sigma^2_{gxg,dist}$ and $\sigma^2_{gxg,local}$ are well-calibrated in simulations (section 'Regional simulation and estimation' of Methods; Supplementary Fig. 12). Applying this test to each of the 16 previously identified marginal epistasis loci, we found 5 and 12 loci with significant local and distal marginal epistasis effects, respectively ($P < \frac{5 \times 10^{-8}}{53}$; Fig. 3b; Supplementary Table 6).

For each of the 16 SNPs, we then ran genome-wide pairwise interaction analysis (GxGWAS) where we tested for the interaction effect of the target SNP on the corresponding trait when paired with remaining SNPs on the UKBB array (after regressing out the same set of covariates as in the marginal epistasis analysis). We found that significant pairwise effects were almost always located on the same chromosome and close to the target SNP (Supplementary Table 7). Of the four trait-SNP pairs demonstrating significant pairwise interaction on the same chromosome, we quantified the distributions of physical distance (Supplementary Fig. 13 (left); mean: 699.3 kb; median: 755.4 kb; min: 5.7 kb; max:1359.9 kb) and LD ($r^2$) (Supplementary Fig. 13 (right), mean: 0.0019; median: 0.0005; min: 0; max: 0.0176) between the target and interactive SNPs. Overall, these SNP pairs are not highly correlated ($r^2 < 0.10$) suggesting that the marginal epistatic effect is unlikely to be driven by LD. The only significant interaction involving SNPs on distinct chromosomes is found in the case of rs72654473 on lipoA. rs72654473, downstream of the *APOE* gene on chromosome 19, shows a significant interaction effect with rs6935921, near the *LPA* gene on chromosome 6, where rs6935921-T allele attenuates the effect of rs72654473 on lipoA (Fig. 5a).

## Magnitude of marginal epistasis effects

We estimated the proportion of trait variance explained by marginal epistasis at a target SNP $t$ (marginal epistasis heritability, $h^2_{gxg,t}$) from the variance components estimated by FAME (Supplementary Note 4). Across the 16 trait-SNP pairs with significant marginal epistasis signal, estimates of $h^2_{gxg,t}$ tend to be modest: $10^{-3} - 10^{-2}$. We compared these estimates to the heritability of the SNP based on its GWAS effect size ($h^2_{gwas,t}$ estimated as the square of the GWAS effect size for a standard-

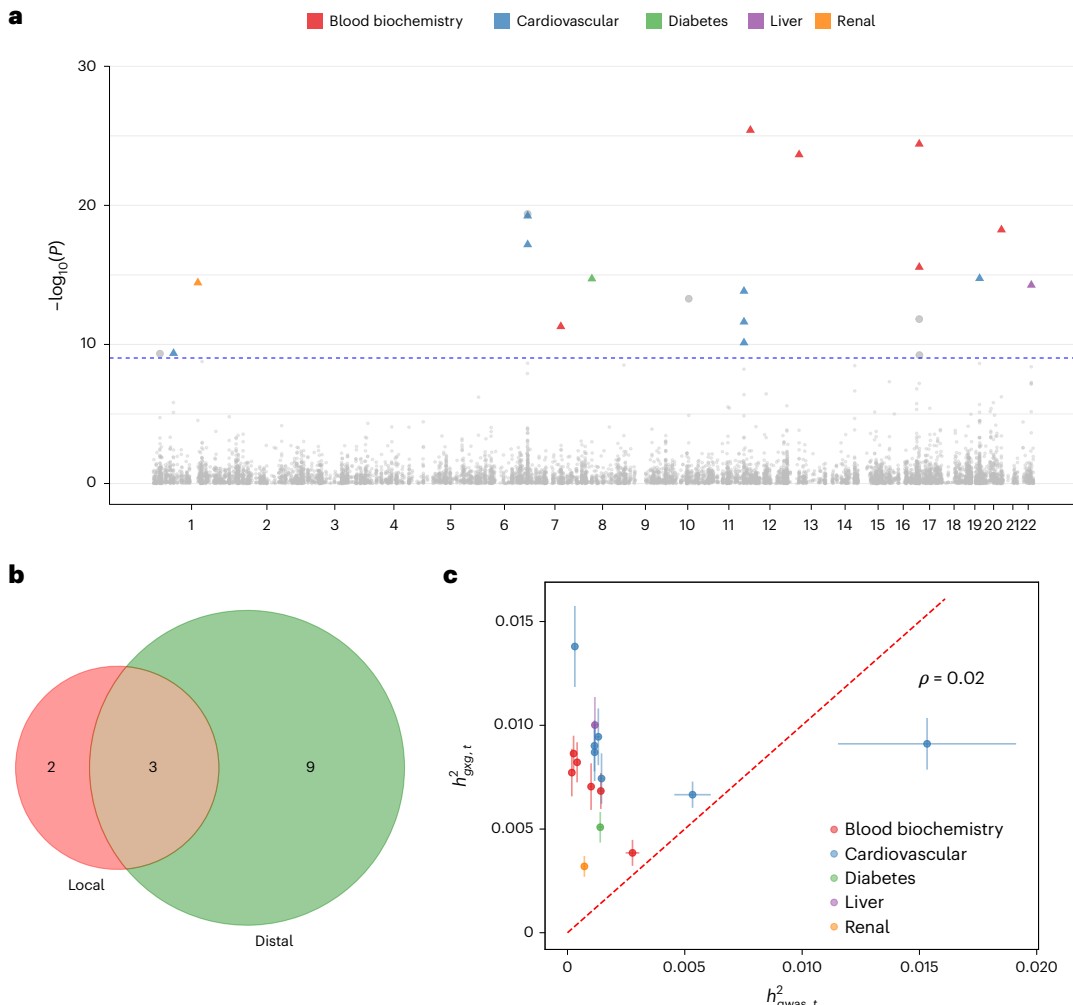

**Fig. 3 | Marginal epistasis signals in the UKBB. a**, Manhattan plot of marginal epistasis loci across 53 complex traits in UKBB. Colored shapes denote trait-SNP pairs that are significant at $P < \frac{5 \times 10^{-8}}{53}$; shapes with colored triangles denote loci that are statistically significant in our initial analysis and remained significant after we regressed out all SNPs within the LD block as fixed effects. **b**, Localization of marginal epistasis signals. For each of 16 trait-SNP pairs, we tested whether the marginal epistasis signals remained significant when testing against all SNPs on the same chromosome as the target SNP (after removing SNPs in the same LD block as the target SNP), which we term local, and against all SNPs on

chromosomes different from the chromosome containing the target SNP, which we term distal. We then compared the overlap between the local and distal significant signals ($P < \frac{5 \times 10^{-8}}{53}$). **c**, The fraction of phenotypic variance explained by marginal epistatic effects ($h^2_{gxg,t}$) vs that explained by additive effects tagged by the genome-wide significant SNP $t$ ($h^2_{gwas,t}$) for each of the 16 marginal epistasis significant trait-SNP pairs. Each dot represents the point estimates. Vertical (horizontal) bars denote the standard error of $h^2_{gxg,t}$ ($h^2_{gwas,t}$). We describe the estimation of $h^2_{gwas,t}$ and its standard error in Supplementary Note 6.

ized genotype). We found that $h^2_{gxg,t}$ estimates are larger than the corresponding $h^2_{gwas,t}$ estimates: about 12$x$ larger on average with a range of 0.59 to 43.89 (Fig. 3c and Table 1). Estimates of $h^2_{gxg,t}$ are not strongly correlated with the $h^2_{gwas,t}$ estimates (Pearson's correlation coefficient $\rho = 0.022$).

### Functional interpretation of marginal epistasis signals

We performed a series of analyses to better understand the biological relevance of trait-SNP pairs showing significant marginal epistasis. Overlapping the 14 unique SNPs significant for marginal epistasis with sets of eQTLs and pQTLs (section 'Datasets' of Methods), we found eleven to regulate gene expression or protein levels with ten identified as *cis*-eQTLs, whereas rs146534110, which harbors a marginal epistasis signal for lipoprotein A, was previously identified as a *cis*-pQTL (Supplementary Table 8).

Next, we analyzed the GxGWAS results across all 16 trait-SNP pairs (excluding SNPs in the LD block of the target SNP and the MHC region).

For each significant trait-SNP pair, we selected the 1,000 SNP pairs with the most significant interaction $P$ values. To assess whether these interacting SNPs have functional relevance, we combined sets of SNPs across all traits and used RegulomeDB[38] to test if the interacting SNPs were enriched for functional annotations at the SNP sites (measured as the fraction of sites with rank score ≥1 f) relative to a background of GWAS variants pooled across all outcomes (repeated 10,000 times). For each background set, we randomly selected the same number of GWAS SNPs as the former set and computed their RegulomeDB functional importance scores. We observed that the analysis revealed significant functional enrichment for the interactive SNPs ($P = 0.033$). These results indicate that the SNPs that interact with the marginal epistasis SNPs are significantly enriched for high-ranking regulatory elements.

To test whether the interactive SNPs are enriched for binding of specific transcription factors (TFs), we applied HOMER[39] and analyzed a ±100 bp window around the top 1,000 interactive SNPs for TF motif enrichments. We observed significant enrichment for multiple known

**Table 1 | Candidate trait-SNP pairs with significant marginal epistasis**

| Trait | SNPID | MAF | $h^2_{gxg,t} \times 0.001$ | $P_{gxg,t}$ | $h^2_{gwas,t} \times 0.001$ | $P_{gwas,t}$ | Ratio |
|---|---|---|---|---|---|---|---|
| Alanine aminotransferase | rs3827385 | 0.18 | 10.01 | $1.35 \times 10^{-13}$ | 1.17 | $9.30 \times 10^{-82}$ | 8.57 |
| Apolipoprotein B | rs964184 | 0.13 | 7.43 | $9.05 \times 10^{-10}$ | 1.45 | $1.82 \times 10^{-89}$ | 5.12 |
| C-reactive protein | rs11208750 | 0.20 | 8.70 | $3.66 \times 10^{-10}$ | 1.16 | $5.17 \times 10^{-73}$ | 7.52 |
| Cholesterol | rs964184 | 0.13 | 9.01 | $2.61 \times 10^{-13}$ | 1.15 | $7.50 \times 10^{-74}$ | 7.84 |
| Hemoglobin A1c | rs34265667 | 0.03 | 5.09 | $5.52 \times 10^{-12}$ | 1.40 | $2.58 \times 10^{-94}$ | 3.64 |
| Lipoprotein A | rs628031 | 0.41 | 13.80 | $1.50 \times 10^{-12}$ | 0.31 | $8.07 \times 10^{-17}$ | 43.94 |
| | rs146534110 | 0.01 | 6.65 | $6.50 \times 10^{-26}$ | 5.33 | $3.77 \times 10^{-259}$ | 1.25 |
| | rs72654473 | 0.11 | 9.45 | $5.01 \times 10^{-12}$ | 1.31 | $3.49 \times 10^{-65}$ | 7.18 |
| Mean platelet volume | rs463312 | 0.05 | 6.83 | $2.66 \times 10^{-15}$ | 1.42 | $1.47 \times 10^{-89}$ | 4.81 |
| Monocyte count | rs79490353 | 0.03 | 3.85 | $8.44 \times 10^{-10}$ | 2.77 | $1.79 \times 10^{-183}$ | 1.39 |
| SHBG | rs11656323 | 0.05 | 8.22 | $1.73 \times 10^{-17}$ | 0.41 | $4.30 \times 10^{-29}$ | 20.02 |
| | rs35985803 | 0.08 | 7.04 | $3.43 \times 10^{-19}$ | 1.01 | $8.92 \times 10^{-69}$ | 7.00 |
| Testosterone | rs11555142 | 0.09 | 7.72 | $1.73 \times 10^{-11}$ | 0.18 | $1.62 \times 10^{-29}$ | 41.92 |
| | rs28990703 | 0.04 | 8.64 | $1.03 \times 10^{-23}$ | 0.26 | $6.11 \times 10^{-41}$ | 33.28 |
| Triglycerides | rs964184 | 0.13 | 9.11 | $2.58 \times 10^{-13}$ | 15.33 | $\leq 1 \times 10^{-259}$ | 0.59 |
| Urate | rs75246752 | 0.01 | 3.20 | $1.62 \times 10^{-10}$ | 0.72 | $1.28 \times 10^{-62}$ | 4.44 |

The 16 GWAS candidate trait-SNP pairs passing the significance threshold $\frac{5 \times 10^{-8}}{53}$ are displayed in this table. For each pair, we report the heritability explained by marginal epistasis effect ($h^2_{gxg,t}$), the $P$ value of the marginal epistasis variance component ($P_{gxg,t}$). We also report the heritability based on the GWAS effect size ($h^2_{gwas,t}$), the GWAS $P$ value ($P_{gwas,t}$), and the ratio ($\frac{h^2_{gxg,t}}{h^2_{gwas,t}}$, labeled as 'Ratio').

**Table 2 | Information on trait-SNP pairs with significant marginal epistasis**

| Trait | SNPID | Effect allele | Chr:Pos (b37) | $\hat{\beta} \pm SE \times 0.001$ | Nearest gene | Variant type |
|---|---|---|---|---|---|---|
| Alanine aminotransferase | rs3827385 | T | 22:44388817 | 62.70±3.27 | *SAMM50* | Intron |
| Apolipoprotein B | rs964184 | C | 11:116648917 | 79.44±3.96 | *ZPR1* | 3' UTR |
| C-reactive protein | rs11208750 | C | 1:66257838 | −60.66±3.36 | *PDE4B* | Regulatory |
| Cholesterol | rs964184 | C | 11:116648917 | 70.62±3.88 | *ZPR1* | 3' UTR |
| Hemoglobin A1c | rs34265667 | G | 8:41542093 | −145.27±7.05 | *ANK1* | Synonymous |
| Lipoprotein A | rs628031 | G | 6:160560845 | 25.38±3.05 | *SLC22A1* | Missense |
| | rs146534110 | G | 6:160578069 | 468.07±13.59 | *SLC22A1* | Intron |
| | rs72654473 | C | 19:45414399 | −83.11±4.87 | *APOE* | Non-coding exon |
| Mean platelet volume | rs463312 | A | 20:57597970 | −124.95±6.22 | *TUBB1* | Missense |
| Monocyte count | rs79490353 | T | 13:28623048 | 236.49±8.18 | *FLT3* | Intron |
| SHBG | rs11656323 | T | 17:7145117 | −66.54±5.94 | *GABARAP* | 5' UTR |
| | rs35985803 | G | 17:7254315 | −81.97±4.68 | *ACAP1* | Missense |
| Testosterone | rs28990703 | A | 12:2977954 | 61.76±4.61 | *FOXM1* | Intron |
| | rs11555142 | G | 7:99032593 | −34.35±3.04 | *ATP5J2-PTCD1* | Synonymous |
| Triglycerides | rs964184 | C | 11:116648917 | 257.75±3.81 | *ZPR1* | 3' UTR |
| Urate | rs75246752 | G | 1:145630111 | −167.92±10.05 | *RNF115* | Intron |

For each significant trait-SNP pair, we provide the GWAS allelic effect size estimate and corresponding standard error (SE), the closest protein-coding gene to the SNP, and the variant type.

(FDR < 0.05) (Supplementary Data 1) and de novo ($P < 1 \times 10^{-12}$) TF binding sites (Supplementary Data 2). Overall, these analyses revealed enrichments especially for TFs with helix-loop-helix domains as well as for GATA and nuclear hormone receptor TFs with zinc finger domains, which are known to be involved in the regulation of metabolic traits[40–42].

**Interpretation of individual marginal epistasis loci**
We then attempted to investigate individual marginal epistatic loci for alanine aminotransferase and testosterone (see Supplementary Note 5 for interpretation of loci underlying lipoprotein A and lipids).

**Alanine aminotransferase.** We investigated rs3827385, which shows significant marginal epistasis for alanine aminotransferase (ALT) and lies in the intron of the *SAMM50* (Table 2), a gene whose protein product plays an important role in mitochondrial structure and function[43] and in fatty acid oxidation[44]. This variant is located approximately 60 kb from and in moderate linkage disequilibrium ($r^2 = 0.36$, Lewontin's $D' = 0.67$) with a well-studied missense variant (rs738409, p.I148M) in the functionally relevant PNPLA3 gene. Previous studies have established the association of rs738409 with chronic liver disease[45,46]. Our analysis revealed significant marginal epistasis effects of rs738409 on

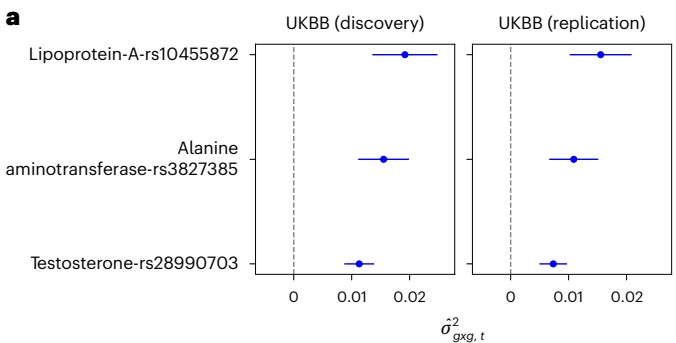

**Fig. 4 | Replicability of the findings from FAME.** We report point estimates (and 95% CIs) of $\sigma^2_{gxg,t}$ for genome-wide significant marginal epistasis loci ($P < \frac{5\times10^{-8}}{53}$) discovered in a discovery cohort and their estimates in a replication cohort. **a**, Our discovery cohort is about one-half of the unrelated White British individuals in UKBB, whereas the replication cohort consists of unrelated White British individuals in UKBB not used in the discovery cohort. **b**, We use the discovery cohort comprising all unrelated White British individuals in UKBB and the replication cohort comprising individuals of European ancestry in the AoU dataset. The detailed sample size of each trait in AoU is reported in Supplementary Table 5.

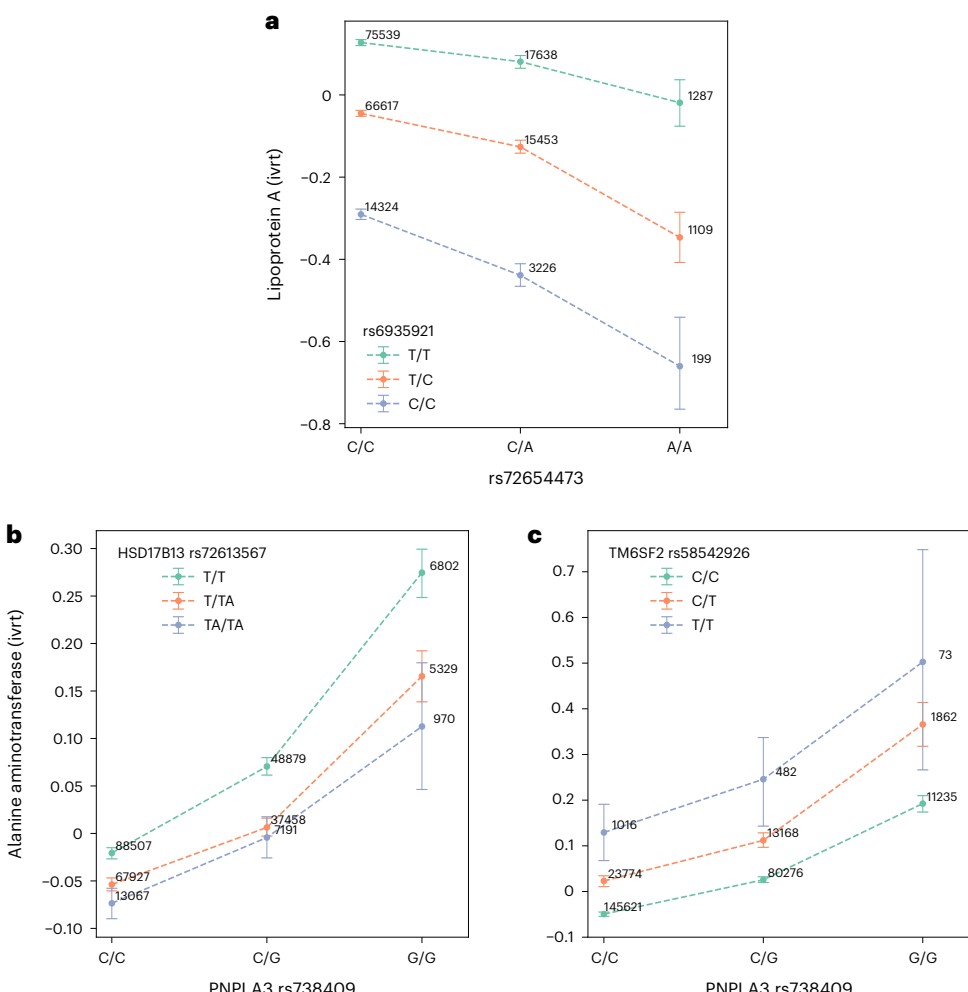

**Fig. 5 | Pairwise epistasis test.** We analyze the relationship between (**a**) rs6935921 and rs72654473 on lipoprotein A; (**b**) rs72613567 and rs738409 and (**c**) rs58542926 and rs738409 on alanine aminotransferase. The error bar represents the 95% confidence interval from 1,000 bootstrap replicates. ivrt, rank-based inverse normal transformation.

ALT, with a proportion of variance explained comparable to that of rs3827385 ($P = 4.90 \times 10^{-24}$, $\sigma^2_{gxg,t} = 14.8 \times 10^{-3}$). Notably, prior work[47] has identified an interaction between rs738409 and a splice variant rs72613567 in *HSD17B13* on chromosome 4, affecting liver function as measured by ALT and aspartate aminotransferase (AST) levels.

Specifically, it has been found that the *HSD17B13* rs72613567:TA allele, associated with reduced aminotransferase levels, attenuates the effect of rs738409 on aminotransferase levels and is linked to reduced *PNPLA3* mRNA expression. A pairwise epistasis test between rs72613567 and rs738409 for ALT in UKBB data confirmed this interaction is nominally

significant ($\hat{\beta}_{pairwise} = -2.52 \times 10^{-2}$, $P = 1.94 \times 10^{-7}$; Supplementary Table 9 and Fig. 5b). Given the established role of rs738409 in liver function, we next sought to investigate additional potential interactors. We selected rs58542926, a missense variant in the gene *TM6SF2*, that has been shown to be associated with ALT and liver disease[48]. We find that the interaction between rs58542926 and rs738409 on ALT is nominally significant with the rs58542926:T allele increasing the effect of rs734809 on ALT levels ($\hat{\beta}_{pairwise} = 2.91 \times 10^{-2}$, $P = 3.48 \times 10^{-4}$; Supplementary Table 9 and Fig. 5c). The proportion of the phenotypic variance explained by the two interactive pairs is less than 1% of the variance explained by marginal epistasis at rs738409 (Supplementary Table 9). Taken together, the increased significance of the marginal epistasis analysis using FAME and the finding that the interactive pairs account for a modest fraction of the marginal epistasis signal indicates the effectiveness of testing the aggregated epistasis effect.

**Testosterone.** Given that testosterone is a highly sex-specific trait, we investigated sex-specificity of marginal epistasis effects of the two testosterone-associated marginal epistasis variants, rs11555142 and rs28990703. We observed differences in marginal epistasis estimates across sexes at both SNPs, with rs11555142 being significant only in females and rs28990703 being significant only in males (Supplementary Fig. 14). Using HaploReg 4.2 (ref. 49), we observed that the SNP rs58964157, which is in complete linkage disequilibrium (LD; $R^2 = 1.0$ in Europeans) with rs11555142, alters the binding of estrogen receptor 2 (*ESR2*) and estrogen receptor α (*ESR1*), the two critical transcription factors (TFs) for estrogen regulation[50,51], consistent with rs11555142 having a significant female-specific marginal epistasis effect. We also identified rs28990703 as a binding site for *LEF1*, a TF known to regulate androgen signaling and implicated in prostate cancer[52]. No tight LD proxies were found for rs28990703, with the closest LD proxy of $R^2 = 0.4$ in HaploReg 4.2. Intriguingly, this site also binds *ZNF652*, a TF linked to prostate cancer and androgen metabolism[53], which are both sex specific. These findings suggest that rs28990703 itself may directly contribute to the observed sex differences by interacting with other variants and influencing androgen-related pathways.

## Discussion

We have presented a new method, FAME, that can detect marginal epistasis in biobank-scale data. A significant challenge to robust identification of marginal epistasis (and epistasis more generally) arises due to model violations. In extensive simulations that include non-linear covariates, gene-environment interactions, heavy-tailed environmental noise, and imperfect tagging of causal variants, we show that FAME yields calibrated results. Applying FAME to 53 quantitative phenotypes in the UK Biobank, we found 16 trait SNP pairs with genome-wide significant signals of marginal epistasis. Although the number of signals detected is modest (in part due to the stringent *P* value threshold employed and the strategy employed to select target SNPs), we observe that the proportion of variance explained by marginal epistasis is comparable to, and sometimes substantially larger than, the proportion of variance explained by GWAS. This observation suggests that polygenic background can substantially modulate the effect of a genetic variant on a trait and has implications for efforts to interpret variant effects, improve phenotype prediction and understand how genetic effects vary across populations[5].

We further partitioned the marginal epistasis signal within and across chromosomes to detect both within and cross-chromosomal signals. Although our current application has focused on testing for interactions of a single target SNP paired with SNPs across the genome, the model underlying FAME is flexible. For example, FAME can be extended to test for interactions of a target SNP or covariate (such as polygenic scores) with SNPs defined based on functional annotations such as genes or pathways.

The scalability of FAME arises from its use of the randomized MoM estimator. Likelihood-based methods that maximize the likelihood or

restricted likelihood (REML) (using the EM, NR, Fisher or AI algorithms for optimization) could be alternatively employed. When effect sizes and the environmental noise are all normally distributed (leading to a tractable likelihood), REML is more statistically efficient than MoM. Most important to our application, the computational efficiency of randomized MoM enables its application to biobank-scale datasets where, as far as we know, likelihood-based approaches are infeasible. The question of which estimator to use depends on considerations of model assumptions and dataset size. Efficient likelihood-based estimators in epistasis models remain an interesting topic for future research.

Our study has several limitations. First, our approach of regressing out additive effects in the LD block surrounding the target SNP and only testing for interactions outside is important for calibration but likely misses local epistatic signals or signals in regions of long-range linkage disequilibrium[54]. Additionally, the model underlying FAME assumes that interaction effects are independent of main effects. It would be of interest to extend our method to settings where epistasis is coordinated[26,55]. Second, the scale on which phenotypes are measured can affect our results. In all our simulations and analyses, traits are inverse rank normalized so that they are invariant to monotone transformations. Although analyzing phenotypes on other scales might lead to more interpretable estimates, the robustness of such analyses needs to be assessed (Supplementary Note 3). Third, we have only applied FAME to quantitative traits. Analyses of binary (disease) traits would require exploring the impact of trait prevalence and ascertainment. Fourth, although we have focused on testing for marginal epistasis at common SNPs, it would be of great interest to apply FAME to rare variants. Fifth, our estimates of marginal epistasis effects are likely to be biased upwards due to winner's curse[56] although our replication experiment suggests that the bias is modest. Sixth, we have limited our analysis to GWAS significant SNPs in this study. The ability to estimate marginal epistasis across the genome would allow us to obtain a more comprehensive understanding of the nature of epistasis underlying complex traits. Finally, integration of FAME with data from biological pathways and functional genomics will be essential to move towards an understanding of biological epistasis[10]. Extending the scope, efficiency and generality of FAME present important directions for future work.

## Online content

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

## Methods

### Ethics approval

Ethics committee/IRB of UKBB gave ethical approval for collection of UKBB data (https://www.ukbiobank.ac.uk/learn-more-about-uk-biobank/about-us/ethics). Approval to use UKBB individual level in this work was obtained under application 33127 at http://www.ukbiobank.ac.uk. Ethics committee/IRB of AoU gave ethical approval for collection of AoU data (https://allofus.nih.gov/about/who-we-are/institutional-review-board-irb-of-all-of-us-research-program). Approval to use AoU controlled tier data in this work was obtained through application at https://www.researchallofus.org.

### Marginal epistasis model

Given an $N \times M$ standardized genotype matrix $X$, an $N$-vector of centered phenotypes $\mathbf{y}$ and a target SNP $t \in \{1, \dots, M\}$ we aim to jointly test the additive effect of the $M$ SNPs and marginal epistasis of the target SNP using the following model that was originally introduced in[25]:

$$\mathbf{y} = X\boldsymbol{\beta} + E_t\boldsymbol{\alpha}_t + \boldsymbol{\varepsilon}$$

$$\boldsymbol{\varepsilon} \sim \mathcal{N}\left(\mathbf{0}, \sigma_e^2 I_N\right)$$

$$\boldsymbol{\beta} \sim \mathcal{N}\left(\mathbf{0}, \frac{\sigma_g^2}{M} I_M\right)$$

$$\boldsymbol{\alpha}_t \sim \mathcal{N}\left(\mathbf{0}, \frac{\sigma_{gxg,t}^2}{M-1} I_{M-1}\right)$$

Here $\mathcal{N}(\boldsymbol{\mu}, \Sigma)$ is a multivariate normal distribution with mean $\boldsymbol{\mu}$ and covariance matrix $\Sigma$, $E_t$ denotes an $N \times (M-1)$ genetic interaction matrix formed by pairing the target SNP with all other SNPs in the genome, defined formally as $E_t = X_{-t} \odot X_{:t}$ where $X_{:t}$ is the $t$-th column (corresponding to the target SNP) of $X$ and $X_{-t}$ is the matrix consisting of all SNPs except the target SNP formed by excluding the $t$-th column from $X$.

In this model, $\sigma_e^2$, $\sigma_g^2$ and $\sigma_{gxg,t}^2$ are the residual, additive genetic and marginal epistasis variance components, respectively. $\boldsymbol{\beta}$ denotes the vector of additive SNP effects and $\boldsymbol{\alpha}_t$ denotes the vector of interaction effects between target SNP $t$ and each of the other SNPs in the genome. This model assumes that the interaction effects are independent of the main effects so that epistasis is uncoordinated[26].

### Estimation of variance components

To estimate the variance components of our LMM, we use a Method-of-Moments (MoM) estimator. As $E[\mathbf{y}] = \mathbf{0}$, we derived the MoM estimates by equating the population covariance to the empirical covariance where the population covariance is given by: $\Sigma = \text{Cov}(\mathbf{y}) = E[\mathbf{y}\mathbf{y}^T] - E[\mathbf{y}]E[\mathbf{y}^T] = \sigma_g^2 \frac{1}{M}XX^T + \sigma_{gxg,t}^2 \frac{1}{M-1}E_tE_t^T + \sigma_e^2 I$. Using $\mathbf{y}\mathbf{y}^T$ as our estimate of the empirical covariance, we need to solve the following least squares problem to estimate the variance parameters:

$$\left(\tilde{\sigma}_g^2, \tilde{\sigma}_{gxg,t}^2, \tilde{\sigma}_e^2\right) = \text{argmin}_{(\sigma_g^2, \sigma_{gxg,t}^2, \sigma_e^2)} ||\mathbf{y}\mathbf{y}^T - (\sigma_g^2 K_1 + \sigma_{gxg,t}^2 K_{2,t} + \sigma_e^2 K_3)||_F^2$$

where $K_1 = \frac{1}{M}XX^T$, $K_{2,t} = \frac{1}{M-1}E_tE_t^T$ and $K_3 = I_N$.

We show that the MoM estimator satisfies the following normal equations (Lemma 1 in Supplementary Notes):

$$T\boldsymbol{\sigma}^2 = \mathbf{q}$$

where $T$ is a $3 \times 3$ matrix with entries $T_{kl} = tr(K_kK_l)$, $k, l \in \{1, 2, 3\}$ $tr()$ denotes the trace of the matrix, and $\mathbf{q}$ is a 3-vector with entries $\mathbf{q}_k = \mathbf{y}^T K_k \mathbf{y}$.

Thus, the MoM estimate of the variance components $\tilde{\boldsymbol{\sigma}}^2 = (\tilde{\sigma}_g^2, \tilde{\sigma}_{gxg,t}^2, \tilde{\sigma}_e^2)$ is given by:

$$\tilde{\boldsymbol{\sigma}}^2 = T^{-1}\mathbf{q}$$

To compute standard errors which, in turn, allow us to test the hypothesis of no marginal epistasis ($\sigma_{gxg,t}^2 = 0$), we compute the covariance matrix of $\tilde{\boldsymbol{\sigma}}^2$ as (Lemma 2 in Supplementary Notes):

$$\text{Cov}[\tilde{\boldsymbol{\sigma}}^2] = T^{-1}\text{Cov}[\mathbf{q}]T^{-1}$$

where

$$\text{Cov}[\mathbf{q}]_{ij} = 2tr(\Sigma K_i \Sigma K_j)$$

### Efficient computation of variance components

Computing the MoM estimates and their standard errors requires $O(N^2 M)$ runtime and $O(N^2)$ memory that render it impractical for biobank-scale data.

To efficiently compute $\tilde{\boldsymbol{\sigma}}^2$, we note that each of the coefficients of the matrix $T$ involves computing the trace of a matrix which we approximate by a stochastic trace estimator[57]. Specifically, we estimate $T_{kl}$ as:

$$\hat{T}_{kl} \approx \frac{1}{BM_kM_l} \sum_{b=1}^{B} \mathbf{v}_b^T Z_k Z_k^T Z_l Z_l^T \mathbf{v}_b$$

where each $\boldsymbol{v}_b$ is an independent random vector with mean zero and covariance $I_N$, $B$ is the total number of random vectors used for the approximation, and $Z_k = X$ or $E_t$. Each term $Z_kZ_k^T\mathbf{v}_b$ can be computed in $O\left(\frac{NM}{\max(\log_3 N, \log_3 M)}\right)$ by using the Mailman algorithm[58] so that $\hat{T}_{kl}$ can be computed in $O\left(\frac{NMB}{\max(\log_3 N, \log_3 M)}\right)$ time.

We note that $\hat{T}_{kl}$ is an unbiased estimator of $T_{kl}$ provided $\mathbf{v}_b$ has zero mean and identity covariance matrix. The distribution of $\mathbf{v}_b$ can impact the variance of the estimator. In practice, we draw each entry of $\mathbf{v}_b$ independently from a standard normal distribution. The variance of $\hat{T}_{kl}$ decreases with the increasing number of random vectors ($B$). Empirically, we observed that utilizing 100 random vectors yields a sufficiently accurate estimator (Supplementary Fig. 9).

To estimate Cov $[\mathbf{q}]$ efficiently, we use a plug-in estimate of $\Sigma$ to obtain:

$$\widehat{\text{Cov}[\mathbf{q}]}_{kl} = 2\mathbf{y}^T K_k \tilde{\Sigma} K_l \mathbf{y} = 2\mathbf{y}^T K_k \left(\sum_{t=1}^{3} \tilde{\sigma}_t^2 K_t\right) K_l \mathbf{y}$$

$$= 2\left(\sum_{t=1}^{3} \tilde{\sigma}_t^2 \mathbf{y}^T K_k K_t K_l \mathbf{y}\right)$$

$$= 2\sum_{t=1}^{3} \tilde{\sigma}_t^2 \left(\mathbf{w}_k^T \frac{Z_t Z_t^T}{M_t} \mathbf{w}_l\right)$$

where $\mathbf{w}_k = K_k\mathbf{y}, k \in \{1, 2, 3\}$.

$\widehat{\text{Cov}[\mathbf{q}]}_{kl}$ can be efficiently computed by writing $\mathbf{w}_k = K_k\mathbf{y}$ as $Z_kZ_k^T\mathbf{y}$ which can be efficiently computed. Further, FAME uses a streaming implementation that does not require all the genotypes to be stored in memory leading to scalable memory requirements. FAME can also account for fixed-effects covariates such as age, sex and genetic PCs (Supplementary Note 7).

We allow for negative variance component estimates that we exclude from the analysis, unless specified otherwise. The corresponding $P$ value is computed using a two-sided test.

### Simulations

**Simulations to assess power and accuracy.** We designed simulations to assess the power of FAME and the accuracy of its marginal epistasis variance components estimates using the following generative model:

$$\mathbf{y} = X\boldsymbol{\beta} + E_t\boldsymbol{\alpha}_t + \boldsymbol{\varepsilon}$$

$$\boldsymbol{\varepsilon} \sim \mathcal{N}(\mathbf{0}, \sigma_e^2 I_N)$$

$$\beta_j \overset{\text{iid}}{\sim} \begin{cases} \mathcal{N}\left(0, \frac{\sigma_g^2}{|M_a|}\right), \text{if } j \in M_a \\ 0, \quad \text{otherwise} \end{cases}$$

$$\alpha_{t,j} \overset{\text{iid}}{\sim} \begin{cases} \mathcal{N}\left(0, \frac{\sigma_{gxg,t}^2}{|M_e|}\right), \text{if } j \in M_e \\ 0, \quad \text{otherwise} \end{cases}$$

Here $\beta_j$ and $\alpha_{t,j}$ denotes the $j$-th element in the respective vectors of effect sizes. We set $\sigma_g^2$ to 0.3, which is approximately the median value of the additive heritability across all the traits that we analyzed in this study. We varied the value of $\sigma_{gxg,t}^2$ from 0.001 to 0.1. We randomly selected 10% of the SNPs to be causal for the additive effects (assigned to the indicator set $M_a$) and 10% of the SNPs to be causal for the marginal epistasis effect (assigned to the indicator set $M_e$, which do not overlap with $M_a$ and fall outside the LD block of the target SNP). As target SNP, we selected three representative SNPs with a range of MAF ($\in\{1\%, 14\%, 49\%\}$, respectively). For computational convenience, we used genotypes in the *UKBB-small* dataset (section 'Datasets' of Methods). We simulated 1,000 replicates for each setting.

To assess the accuracy of the marginal epistasis variance component estimates obtained by FAME, we used the same simulations as above. We then assumed that the target SNPs are known and then estimated the marginal epistasis effect by partitioning the SNPs in $X$ into two bins; the first contains all the SNPs within the same LD block as the target SNP, whereas the second contains all SNPs outside of the block. We used FAME to jointly fit the additive effect for both regions while only fitting the marginal epistasis effect on the second region.

**Regional simulation and estimation.** To localize the marginal epistasis signal, we partitioned the whole genome into the region with all the SNPs lying in the same chromosome as the target SNP but outside of the LD block (termed local) and all the SNPs lying on chromosomes different from the one with the target SNP (termed distal). To validate the calibration of FAME when applied to test the marginal epistasis effect on a specified region, we used the simulation with a total heritability of 0.25 and ratio of causal SNPs of 1%. We applied FAME to estimate the calibration of $\sigma_{gxg,local}^2$ and $\sigma_{gxg,distal}^2$, respectively.

## Runtime comparisons

All experiments used a machine equipped with AMD EPYC 7501 32-Core Processor, and a runtime budget of 3 days was provided to all tested methods.

## Datasets

**Simulation dataset.** We obtained a set of $N = 291{,}273$ unrelated White British individuals measured at $M = 459{,}792$ common SNPs genotyped on the UK Biobank Axiom array to use in simulations by extracting individuals that are more distantly related than third-degree relatives and excluding individuals with putative sex chromosome aneuploidy. Unless otherwise specified, all simulations were conducted using this dataset.

**UKBB-small.** We used a subset of the UKBB array genotypes obtained by restricting to SNPs on chromosomes 12 and 20 across 291,273 unrelated White British individuals. We excluded SNPs with MAF less than 1% resulting in a final set of 32,708 SNPs. We term this dataset 'UKBB-small'. For simulations that involve missing SNPs, we constructed imputed genotypes by restricting to SNPs on chromosomes 12 and 20 named 'UKBB-small-imputed'.

**UKBB genotypes.** For analysis of real traits, we restricted our analysis to SNPs that were present in the UK Biobank Axiom array used to genotype the UK Biobank. SNPs with greater than 1% missingness and minor allele frequency smaller than 1% were removed. Moreover, SNPs that fail the Hardy–Weinberg test at significance threshold $10^{-7}$ were removed. We restricted our study to self-reported British White ancestry individuals who are more than third-degree relatives, defined as pairs of individuals with kinship coefficient $<1/2^{(9/2)}$ (ref. 28). Furthermore, we removed individuals who are outliers for genotype heterozygosity and/or missingness and excluded SNPs that fall within the MHC region. Finally, we obtained a set of $N = 291{,}273$ individuals and $M = 454{,}207$ SNPs for real data analyses. We used this dataset in our analyses unless specified otherwise.

We also analyzed imputed genotypes across $N = 291{,}273$ unrelated White British individuals. We removed SNPs with greater than 1% missingness, minor allele frequency smaller than 1%, SNPs that fail the Hardy–Weinberg test at significance threshold $10^{-7}$ as well as SNPs that lie within the MHC region (Chr6:25–35 Mb) to obtain 4,824,392 SNPs.

**Covariates and phenotypes.** We selected 53 quantitative traits in UKBB (Supplementary Table 10). These traits were chosen because they have been analyzed in prior work, by us and others, to estimate aspects of genetic architecture[59,60] and span a diverse set of phenotypic categories: Anthropometry, Blood Biochemistry, Bone, Cardiovascular, Diabetes, Eye, Liver, and Renal. We included sex, age and the top 20 genetic PCs as covariates in our analysis for all phenotypes. Extra covariates were added for diastolic/systolic blood pressure (adjusted for cholesterol-lowering medication, blood pressure medication, insulin, hormone replacement therapy and oral contraceptives). We used the PCs computed in the UKBB from a superset of 488,295 individuals. Following prior studies, all traits were inverse rank normalized[60].

**UKBB GWAS.** We ran GWAS on each of the 53 traits, including covariates as described above. For each trait, we selected SNPs with $P < 5 \times 10^{-8}$ followed by LD pruning (using a window size of 500 SNPs, we computed $r^2$ between each pair and removed one of them if $r^2 > 0.1$, shifting the window by 1 SNP, and repeating the process).

**AoU dataset.** We curated the AoU genotype dataset starting from the AoU srWGS dataset and identifying the subset of SNPs that matched the UKBB SNP set resulting in 457,218/459,792 (99.4%) SNPs. We restricted our analyses to individuals whose European ancestry component of genetically inferred ancestry (GIA) > 95%. We then applied a genotype QC procedure with the same parameters used for the UKBB yielding 392,756 SNPs. For each trait, when multiple measurements were available, we selected the earliest measurement and adjusted the age covariate accordingly for that individual. Individuals whose adjusted age was <18 years were excluded from the analysis. For the 12 traits that showed evidence for genome-wide marginal epistasis signals, we found five with >50 000 individuals with European ancestry. We used the top five PCs, age and sex as covariates following the recommendations of the AoU investigators[29].

**pQTL and eQTL datasets.** We analyzed associations that were discovered in the following three datasets:

1. Võsa et al. identified *cis*-eQTLs (SNP-gene distance<1 Mb, FDR < 0.05) for 16,987 genes and *trans*-eQTLs (SNP-gene distance>5 Mb, FDR < 0.05) for 6,298 genes in 31,684 blood samples[61]. For *trans*-eQTL analyses, this study focused on 10,317 trait-associated SNPs from GWAS.

2. Yao et al. measured the levels of 71 plasma proteins that are known to be associated with cardiovascular disease (CVD) in 6,861 Framingham Heart Study (FHS) participants. We analyzed the 16,602 protein QTL (pQTL) variants associated with 57 proteins identified in this study[62]. This set included 11,806 cis-pQTL variants ($P < 1.25 \times 10^{-7}$) for 40 proteins and 4,796 trans-pQTL variants ($P < 7.04 \times 10^{-10}$) for 44 proteins.

3. Ferkingstad et al. measured the levels of 4,907 plasma proteins in 35,559 Icelandic individuals. We analyzed the 18,084 sentinel pQTL associations (defined as the most significant association detected within a 1 Mb region surrounding the gene encoding the protein) associated with 4,631 proteins identified in this study ($P < 1.8 \times 10^{-9}$ corresponding to an FDR of 1.3%; ref. 63). Of the sentinel pQTL associations, 1,881 were *cis* whereas 16,203 were *trans*.

## Reporting summary

Further information on research design is available in the Nature Portfolio Reporting Summary linked to this article.

## Data availability

The UK Biobank dataset used in this study is not publicly available but can be obtained by application (https://www.ukbiobank.ac.uk/). The All of Us dataset used in this study can be accessed via the public Data Browser upon approval (https://databrowser.researchallofus.org/). The eQTL data analyzed in this study are available from the eQTLGen consortium[61]. The pQTL data were obtained from the supplementary materials of Yao et al.[62] (Supplementary Table 3) and Ferkingstad et al.[63] (Supplementary Table 2).

## Code availability

FAME can be found at https://github.com/sriramlab/FAME and has been archived on Zenodo at https://doi.org/10.5281/zenodo.15814715 ref. 64 The simulator used in the experiments can be found at https://github.com/sriramlab/Simulator. MAPIT can be found at https://github.com/lorinanthony/MAPIT.

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

## Acknowledgements

This research was conducted using the UK Biobank Resource under application 33127. We gratefully acknowledge the participants of UK Biobank and All of Us for their contributions, without whom this research would not have been possible. We also thank the National Institutes of Health's All of Us Research Program for making available the participant data examined in this study. This work was supported, in part, by NIH grants GM125055, GM153406 (B.F., A.P., A.A., P.A., Z.L. and S.S) and HG006399 (S.S.), UCLA DYA (B.F.), and NSF grant CAREER-1943497 (B.F., A.P. and S.S.). N.Z. was supported by NIH grants R01MH130581, U01MH126798, R01MH122688 and R01GM142112. P.P. was supported by NIH grants R01HL170604 and R01DK132775.

## Author contributions

S.S. proposed the idea and supervised the project. B.F., A.P. and S.S. contributed to method and software development. B.F. led the experimental design and performed the analyses. B.F. and Z.S. contributed to the AoU analysis. B.F., A.K., R.B. and P.P. contributed to the biological interpretation. A.X., A.A., P.A. and Z.L. assisted with the experimental analysis. R.B. and N.Z. provided experimental guidance and manuscript proofreading. B.F. and S.S. wrote the paper with the participation of all authors.

## Competing interests

The authors declare no competing interests.

## Additional information

**Correspondence and requests for materials** should be addressed to Sriram Sankararaman.

# Reporting Summary

## Statistics

For all statistical analyses, confirm that the following items are present in the figure legend, table legend, main text, or Methods section.

| n/a | Confirmed | |
|---|---|---|
| ☐ | ☒ | The exact sample size (*n*) for each experimental group/condition, given as a discrete number and unit of measurement |
| ☒ | ☐ | A statement on whether measurements were taken from distinct samples or whether the same sample was measured repeatedly |
| ☐ | ☒ | The statistical test(s) used AND whether they are one- or two-sided *Only common tests should be described solely by name; describe more complex techniques in the Methods section.* |
| ☐ | ☒ | A description of all covariates tested |
| ☐ | ☒ | A description of any assumptions or corrections, such as tests of normality and adjustment for multiple comparisons |
| ☐ | ☒ | A full description of the statistical parameters including central tendency (e.g. means) or other basic estimates (e.g. regression coefficient) AND variation (e.g. standard deviation) or associated estimates of uncertainty (e.g. confidence intervals) |
| ☐ | ☒ | For null hypothesis testing, the test statistic (e.g. *F*, *t*, *r*) with confidence intervals, effect sizes, degrees of freedom and *P* value noted *Give P values as exact values whenever suitable.* |
| ☒ | ☐ | For Bayesian analysis, information on the choice of priors and Markov chain Monte Carlo settings |
| ☐ | ☒ | For hierarchical and complex designs, identification of the appropriate level for tests and full reporting of outcomes |
| ☐ | ☒ | Estimates of effect sizes (e.g. Cohen's *d*, Pearson's *r*), indicating how they were calculated |

*Our web collection on statistics for biologists contains articles on many of the points above.*

## Software and code

Policy information about availability of computer code

| Data collection | No new data was collected for this study. |
|---|---|
| Data analysis | The code for marginal epistasis testing and estimation (FAME) is freely available at https://github.com/sriramlab/FAME . The simulator used to benchmark FAME is freely available at https://github.com/sriramlab/Simulator. MAPIT software can be found at https://github.com/lorinanthony/MAPIT. |

For manuscripts utilizing custom algorithms or software that are central to the research but not yet described in published literature, software must be made available to editors and reviewers. We strongly encourage code deposition in a community repository (e.g. GitHub). See the Nature Portfolio guidelines for submitting code & software for further information.

## Data

Policy information about availability of data

All manuscripts must include a data availability statement. This statement should provide the following information, where applicable:
- Accession codes, unique identifiers, or web links for publicly available datasets
- A description of any restrictions on data availability
- For clinical datasets or third party data, please ensure that the statement adheres to our policy

The UK Biobank dataset used in this study is not publicly available but can be obtained by application (https://www.ukbiobank.ac.uk/). The UK Biobank data was accessed under application number 33127. The All of Us dataset used in this study can be accessed via the public Data Browser upon approval (https://

## Research involving human participants, their data, or biological material

Policy information about studies with [human participants or human data](). See also policy information about [sex, gender (identity/presentation), and sexual orientation]() and [race, ethnicity and racism]().

| | |
|---|---|
| Reporting on sex and gender | We analyzed unrelated white British individuals from the UK Biobank data. We jointly analyzed males and females while regressing out sex as a covariate in our analyses. The majority of our findings are relevant to the general population in the UK Biobank and are not gender/sex specific. We performed sex-specific analysis of marginal espistasis effects underlying serum testosterone. |
| Reporting on race, ethnicity, or other socially relevant groupings | Our analyses focus on white British individuals as defined in the UK Biobank. |
| Population characteristics | The UK Biobank recruited individuals from the UK aged between 49 and 60. Our analysis focused on the subset of unrelated white British individuals in the UK Biobank. All of Us recruited individuals 18 years of age or older from a network of recruitment sites across the USA. |
| Recruitment | No new data was collected for this study. |
| Ethics oversight | Ethics committee/IRB of UKBB gave ethical approval for collection of UKBB data (https://www.ukbiobank. ac.uk/learn-more-about-uk-biobank/about-us/ethics). Approval to use UKBB individual level in this work was obtained under application 33127 at http://www.ukbiobank.ac.uk. Ethics committee/IRB of AoU gave ethical approval for collection of AoU data (https://allofus.nih.gov/about/who-we-are/ institutional-review-board-irb-of-all-of-us-research-program). Approval to use AoU controlled tier data in this work was obtained through application at https://www.researchallofus.org. |

Note that full information on the approval of the study protocol must also be provided in the manuscript.

# Field-specific reporting

Please select the one below that is the best fit for your research. If you are not sure, read the appropriate sections before making your selection.

☒ Life sciences   ☐ Behavioural & social sciences   ☐ Ecological, evolutionary & environmental sciences

For a reference copy of the document with all sections, see [nature.com/documents/nr-reporting-summary-flat.pdf]()

# Life sciences study design

All studies must disclose on these points even when the disclosure is negative.

| | |
|---|---|
| Sample size | For analysis of real traits, we restricted our analysis to SNPs that were present in the UK Biobank Axiom array used to genotype the UK Biobank. SNPs with greater than 1% missingness and minor allele frequency smaller than 1% were removed. Moreover, SNPs that fail the Hardy-Weinberg test at significance threshold $10^{-7}$ were removed. We restricted our study to self-reported British white ancestry individuals which are > 3rd degree relatives that is defined as pairs of individuals with kinship coefficient < 1/2(9/2) [5]. Furthermore, we removed individuals who are outliers for genotype heterozygosity and/or missingness and excluded SNPs that fall within the MHC region. Finally, we obtained a set of N = 291,273 individuals and M = 454,207 SNPs for real data analyses. We used this dataset in our analyses unless specified otherwise. |
| Data exclusions | We excluded individuals in the UK Biobank who were not classified as White British and removed closely related individuals (closer than third-degree relatives). These exclusions were made to reduce potential confounding due to population stratification and other unobserved factors correlated with population structure. |
| Replication | We performed an internal replication within UK Biobank and also an external replication of the UK Biobank results in the All of Us (AoU) where we analyzed traits that had a N >50K individuals of European ancestry in the AoU data. |
| Randomization | Randomization was not applicable, as this study analyzed existing observational data. |
| Blinding | Blinding was not applicable, as this study analyzed existing observational data and did not generate new experimental results. |

# Reporting for specific materials, systems and methods

We require information from authors about some types of materials, experimental systems and methods used in many studies. Here, indicate whether each material, system or method listed is relevant to your study. If you are not sure if a list item applies to your research, read the appropriate section before selecting a response.

## Materials & experimental systems

| n/a | Involved in the study |
|-----|----------------------|
| ☒ ☐ | Antibodies |
| ☒ ☐ | Eukaryotic cell lines |
| ☒ ☐ | Palaeontology and archaeology |
| ☒ ☐ | Animals and other organisms |
| ☒ ☐ | Clinical data |
| ☒ ☐ | Dual use research of concern |
| ☒ ☐ | Plants |

## Methods

| n/a | Involved in the study |
|-----|----------------------|
| ☒ ☐ | ChIP-seq |
| ☒ ☐ | Flow cytometry |
| ☒ ☐ | MRI-based neuroimaging |

## Plants

| | |
|---|---|
| Seed stocks | *Report on the source of all seed stocks or other plant material used. If applicable, state the seed stock centre and catalogue number. If plant specimens were collected from the field, describe the collection location, date and sampling procedures.* |
| Novel plant genotypes | *Describe the methods by which all novel plant genotypes were produced. This includes those generated by transgenic approaches, gene editing, chemical/radiation-based mutagenesis and hybridization. For transgenic lines, describe the transformation method, the number of independent lines analyzed and the generation upon which experiments were performed. For gene-edited lines, describe the editor used, the endogenous sequence targeted for editing, the targeting guide RNA sequence (if applicable) and how the editor was applied.* |
| Authentication | *Describe any authentication procedures for each seed stock used or novel genotype generated. Describe any experiments used to assess the effect of a mutation and, where applicable, how potential secondary effects (e.g. second site T-DNA insertions, mosiacism, off-target gene editing) were examined.* |

