## [Peer Review File · Nature Genetics]

A biobank-scale test of marginal epistasis reveals genome-wide signals of polygenic interaction effects

Corresponding Author: Professor Sriram Sankararaman

A version of this paper was originally rejected for publication by Nature Genetics, however that decision was reconsidered after appeal by the authors.

Version 0:

Decision Letter:

7th September 2023

Dear Sriram,

Thank you for inquiring about our interest in this proposed submission. It sounds interesting and we would like to see it, although I am sure you will appreciate that it can be very difficult to judge a paper without having seen it in its entirety. We cannot guarantee that we will send it out for review until we have had a chance to look at the full manuscript.

Please note that supplementary materials must not be included with the main text and figures of your manuscript. Supplementary items should be uploaded as separate files.

At this stage you will also need to upload:

1) The Editorial Policy Checklist:
<https://www.nature.com/documents/nr-editorial-policy-checklist.pdf>

2) The Reporting Summary:
<https://www.nature.com/documents/nr-reporting-summary.pdf>

(Here you can read about the role of the Reporting Summary in reproducible science:
<https://www.nature.com/news/announcement-towards-greater-reproducibility-for-life-sciences-research-in-nature-1.22062>)

In order to submit your complete manuscript, please use the link below and follow the prompt to “Revise NG-PI63391”:

Link Redacted

We look forward to receiving it.

Sincerely,
Kyle

Kyle Vogan, PhD
Senior Editor
Nature Genetics
<https://orcid.org/0000-0001-9565-9665>

Version 1:

Decision Letter:

28th November 2023

Dear Sriram,

Your Article entitled "A Biobank-scale test of marginal epistasis reveals genome-wide signals of polygenic epistasis" has been seen by three referees, whose comments are included below. In the light of their advice, we have decided that we cannot offer to publish your manuscript in Nature Genetics.

In particular, while the referees find your work of some interest, they raise overlapping technical concerns that limit the strength of the novel conclusions that can be drawn at this stage. We are persuaded that these reservations are sufficiently important as to preclude publication of this study in Nature Genetics.

I am sorry we cannot be more positive on this occasion, but we hope you will find our referees' comments helpful when preparing your paper for submission elsewhere.

Sincerely,
Kyle

Kyle Vogan, PhD
Senior Editor
Nature Genetics
<https://orcid.org/0000-0001-9565-9665>

Referee expertise:

Referee #1: Genetics, complex traits, statistical methods

Referee #2: Genetics, complex traits, statistical methods

Referee #3: Genetics, complex traits, statistical methods

Reviewers' Comments:

Reviewer #1:
Remarks to the Author:

-This is an interesting paper which purports to detect epistatic interactions for several traits within the UK Biobank

Major comments:

-As far as I can see, the method the authors have employed is not particularly novel, basically taking an existing method and then restricting the search space to only those interactions involving a genome-wide significant SNP.

-The authors lack a replication cohort, so it's difficult to know whether the results are "real".

-If we assume that the results are real, they lack biological insight above that provided by knowledge of the main effects. Would it be possible to follow up cases of interaction with 2x2 tests of epistasis to try and localize the interaction?

-Why not evaluate the significance of the tests using permutation i.e. permuting first the phenotype with respect to the genetic data, and then permuting the phenotype and genome-wide significant variant with respect to the rest of the genotype data?

Minor comments:

Introduction:

As far as I'm aware, the first studies to look at the feasibility of performing an exhaustive pairwise search for epistasis in GWAS data are Marchini et al. 2005 Nat Genet and Evans et al. 2006 PLoS Genet. These should both be cited.

Results:

- "...are the residual variance, genetic variance and the ME variance". Perhaps specify "additive genetic variance" to make this explicit for the reader.

-Make clear the matrix symbol is the Hadamard product

-Is there some inflation of the test statistic in the bottom row of Figure S2?

-The final paragraph of section 2.2. My memory of this is that the inflation is only an issue when one of the contributing SNPs has a massive effect. Did the authors' simulate a large main effect when evaluating the method's robustness to this?

-Section 2.3: Specify what the sample size is.

-Section 2.4: Can the authors give some indication of the memory requirements?

-Section 2.5.5 "has been association"

Methods:

-Could the authors please list the 53 traits analyzed in a supplementary table? (apologies if I have missed this). Why were these particular traits chosen?

Figure 2b - what is the sample size? Annotate this in the legend?

Table 1 - Please insert the marginal results in this table including the MAF of the SNP, its (fixed) effect size and p value. Are these known hits? What are the closest genes? Also please list the rs numbers of that variants, not just the base pair position (which build?). Similar comments for Table 2.

Table S2: g count- are these independent GWS SNPs? Please indicate

Table S3: Same comments as above with respect to listing the rs numbers, closest genes, etc.

Limitations:

-Presumably the method will not detect interactions that do not produce large main effects. Perhaps this should be commented on.

-Is the method limited to quantitative traits? Again, perhaps commented on.

Reviewer #2:

Remarks to the Author:

The paper presents a study on marginal epistasis (ME) at a biobank scale, revealing significant genome-wide signals of polygenic epistasis. The authors have utilized a large dataset from the UK Biobank and developed a method to efficiently estimate the covariance matrix for ME. They have analyzed the heritability at loci with significant ME effects, comparing these with heritability estimates based on GWAS effects. Overall, the paper presents significant findings in the field of genetic epidemiology, particularly in understanding polygenic epistasis. However, addressing the below points would greatly enhance the study's clarity, robustness, and impact.

1. The authors exclude SNPs within the LD block around the target SNP when constructing the set of genetic interactions but retain these SNPs in the additive component. This approach might overlook epistatic interactions in long-range linkage disequilibrium (LD) regions (Singhal et al, AJHG 2023). The authors should provide more context on how this method effectively corrects for inflation in these scenarios.

2. Supplementary Figure S2 indicates residual inflation in the Causal Ratio 0.1 scenario. The paper should address how this inflation is corrected or managed, ensuring the robustness of the findings.

3. Based on Supplementary Figure S5, the method appears underpowered for rare variants. The paper should discuss whether this underpowered nature extends to the detection of recessive effects of variants, especially when SNPs are coded for non-additive effects.

4. The heritability estimates from gene-by-gene interactions (g×g) are reported to be 12 times larger than those from GWAS, despite using GWAS significant SNPs. The paper should justify why this significant difference is not a result of false positives.

5. The authors claim that the epistatic signal detected is likely polygenic and that their approach of testing aggregate effects is more powerful. However, this is tested only with GWAS significant variants. The paper should explore whether the same pattern is observed with non-significant variants to strengthen the argument.

6. The study does not include a replication dataset. For the validity and generalizability of the findings, it is crucial to replicate the results in an independent cohort.

7. The study is limited to quantitative traits. The paper should discuss the applicability of the findings to binary traits or

include an analysis of such traits for a more comprehensive understanding.

8. The paper should clarify how SNP p-values, corrected for marginal effects of other SNPs, relate to biological epistasis. For instance, the association with ApoB, Cholesterol, and TG, and the mediation analyses with other genes, should be contextualized in terms of these corrections. This clarification will aid in understanding the biological implications of the statistical findings.

9. The term "spearman correlation" is misspelled as "spearsman" in Figure S6. This should be corrected for accuracy.

Reviewer #3:

Remarks to the Author:

A biobank-scale test of marginal epistasis reveals genome-wide signals of polygenic epistasis

Fu et al.

The study introduces a method to estimate SNP-by-genetic background interactions, and investigates the contribution of epistasis to complex human traits. Existing methods aimed at identifying interactions among genetic variants face challenges such as low power and computational issues due to the testing of numerous hypotheses. Additionally, spurious epistasis signals can arise from untagged causal variants, posing a significant concern. The proposed approach appears to effectively address these limitations. Applied to biobank-scale data, the method's calibrated performance is confirmed through simulations. In the examination of 15,601 trait-loci associations in the UK Biobank, the study successfully identifies 16 trait-loci pairs across 12 traits, uncovering robust evidence of marginal epistasis (ME). These results, the first of their kind, underscore the substantial contribution of ME to trait variance, surpassing the impact of Genome-Wide Association Study (GWAS) effects.

While this study is both important and intriguing, there are several comments that should be clearly addressed.

<Comments>

The accuracy of estimating variance components using the Method of Moments (MoM) can be influenced by several factors, and the use of random variables (B) and simplifications may impact the precision of the estimates. There are several comments on this approach:

1. The choice of these random variables and their characteristics (e.g., distribution, independence) can affect the accuracy and variability of the estimates. Please clearly state the assumptions made for these random variables in the approach. Additionally, elaborate more on this approach in the Results section to help readers understand it better.
2. Ad-hoc simplifications in the estimation procedure may reduce the complexity of calculations. However, these simplifications should be carefully justified to ensure they do not introduce bias or compromise the validity of the estimation. It is crucial to validate MoM estimates against known values or alternative methods where possible (e.g., EM, NR, Fisher, or AI algorithms). Using various scenarios and considering assumptions in the MoM approach, the authors should report situations in which MoM estimates deviate from those obtained through exact methods (EM, NR, etc.) so that they can advise readers on when to use MoM and when not to.
3. When assessing the performance of FAME, showing just a QQ plot may not be sufficient to demonstrate if the type I error rate is genuinely controlled under the null. Please report the genomic inflation factor (aka lambda) and explicitly state the type 1 error rate for all simulation scenarios under the null. Consider including other relevant metrics as well.

The simulation study has some limitations. Please consider the following:

4. The authors should introduce a random variable that interacts with genome-wide SNPs but assume the random variable is unknown in the analysis. In this situation, FAME should be applied in the absence of epistasis, and the authors should assess if the type 1 error rate is controlled. The interaction due to the unknown random variable can be significant, so it should be tested in the range of 0.1 – 0.5 in terms of the phenotypic variance explained by the interaction due to unknown random variables.
5. The authors should consider phenotypic or residual heteroscedasticity, such as heterogeneous residual variances across age, sex, or any other random variables that can be known or unknown. In this situation, FAME should be applied to assess the type 1 error rate as well as the unbiasedness of the estimates.
6. The authors should also introduce large quadratic or polynomial effects of covariates that are not necessarily known. In this situation, FAME should be applied to assess the type 1 error rate as well as the unbiasedness of the estimates.

Additionally, the authors should consider more comprehensive simulation scenarios to ensure the robustness of their findings and mitigate the possibility of spurious results.

Additional comments are as follows:

7. What is the usual range of the magnitude of GxG epistasis? A magnitude of 0.1 for a single genetic variant seems very large. Some references supporting this would be beneficial.

8. The authors should discuss other similar approaches. For instance, a previous study employed the Hadamard product between two genomic relationship matrices to identify interactions between two sets of SNPs:

"Efficient Algorithms for Calculating Epistatic Genomic Relationship Matrices. *Genetics*, Volume 216, Issue 3, 1 November 2020, Pages 651–669, <https://doi.org/10.1534/genetics.120.303459>."

Similarly, other studies have used Cholesky decomposition of two genomic relationship matrices (e.g., regulatory vs. DHS genomic regions) to investigate if their effects are positively or negatively correlated, indicating some interaction between two genomic regions:

"CORE GREML for estimating covariance between random effects in linear mixed models for complex trait analyses. *Nature Communications* volume 11, Article number: 4208 (2020)."

All these studies employed a linear mixed model, similar to the context of this study.

9. In line 136, it is stated, "We observed that FAME remains calibrated indicating its robustness to imperfect tagging (Supplementary Figure S3)." As mentioned in #3 above, please explicitly report the type 1 error rate, genomic inflation factor, and the distribution of estimated GxG. Additionally, as per comment #4, assign large interaction effects on untagged SNP(s) to observe how the type 1 error rate is affected.

10. It is not clear how the authors simulated the data under imperfect tagging scenarios. The authors should also consider untagged haplotype scenarios, as pointed out in reference #45.

11. Regarding the scale of y in line 211, it is mentioned, "We noticed that by changing the scale of the target trait, the p -values from FAME were significantly inflated (Supplementary Figure S9), thus suggesting that the ME signals discovered by FAME are unlikely to arise due to the scale on which traits are measured." However, this does not necessarily rule out the possibility of inflation due to scale, as it depends on the level of the scale. Could the authors explore other transformations (e.g., $y^{1.1}$, ...) to observe how the genomic inflation factor changes? This experimentation can also be conducted through simulations to quantify alterations in the type 1 error rate. While this is related to question #5, it should be separately and explicitly tested.

**Although we cannot offer to publish your manuscript in *Nature Genetics* given these reviews, we suggest that you consider *Nature Communications* as a potential venue for this work after appropriate revisions. To transfer your manuscript to *Nature Communications* or to another journal of your choice within the Nature portfolio, please use our manuscript transfer portal. You will not have to re-supply manuscript metadata and files, unless you wish to make modifications. For more information, please see our [manuscript transfer FAQ](http://www.nature.com/authors/author_resources/transfer_manuscripts.html?WT.mc_id=EMI_NPG_1511_AUTHORTRANSF&WT.ec_id=AUTHOR) page.

Version 2:

Decision Letter:

IMPORTANT: Please note the reference number: NG-A63391R1-Z Sankararaman. This number must be quoted whenever you communicate with us regarding this paper.

6th September 2024

Dear Sriram,

Thank you for asking us to consider a potential resubmission of your manuscript entitled "A Biobank-scale test of marginal epistasis reveals genome-wide signals of polygenic epistasis". I have now discussed proposed revision plan with my editorial colleagues, and we would like to invite you to revise your manuscript along the lines that you propose for editorial consideration and further peer review.

When preparing a revision, please ensure that it fully complies with our editorial requirements for format and style; details can be found in the Guide to Authors on our website (<http://www.nature.com/ng/>).

Please also be sure that your manuscript is accompanied by a separate letter detailing the changes you have made and your response to the points raised. At this stage, we will need you to upload:

1) A copy of the manuscript in MS Word .docx format.

2) The Editorial Policy Checklist:
<https://www.nature.com/documents/nr-editorial-policy-checklist.pdf>

3) The Reporting Summary:
<https://www.nature.com/documents/nr-reporting-summary.pdf>

(Here you can read about the role of the Reporting Summary in reproducible science:
<https://www.nature.com/news/announcement-towards-greater-reproducibility-for-life-sciences-research-in-nature-1.22062>)

Please use the link below to be taken directly to the site and view and revise your manuscript:

Link Redacted

With best wishes,
Kyle

Kyle Vogan, PhD
Senior Editor
Nature Genetics
<https://orcid.org/0000-0001-9565-9665>

Version 3:

Decision Letter:

9th January 2025

Dear Sriram,

Your revised Article "A biobank-scale test of marginal epistasis reveals genome-wide signals of polygenic epistasis" has been seen by the original referees. As you will see from their comments below, they are broadly satisfied with the responses to their previous concerns pending clarification of a few points. (Reviewer #3 provided only positive comments to the editors at this round of review.) We are interested in the possibility of publishing your study in Nature Genetics, but we would like to see your response to these points in the form of a further revision before we make a final decision on publication.

We therefore invite you to further revise your manuscript taking into account all reviewer comments. Please highlight all changes in the manuscript text file. At this stage, we will need you to upload a copy of the manuscript in MS Word .docx or similar editable format.

*2) If you have not done so already, please begin to revise your manuscript so that it conforms to our Article format instructions, available

http://www.nature.com/ng/authors/article_types/index.html here

*3) Include a revised version of any required Reporting Summary: <https://www.nature.com/documents/nr-reporting-summary.pdf>

Please be aware of our <https://www.nature.com/nature-research/editorial-policies/image-integrity> guidelines on digital image standards.

EXTENDED DATA FIGURES

When re-submitting your manuscript, please ensure that any supplementary figures and tables that are crucial to the

manuscript's conclusions are converted into Extended Data figures and tables to increase visibility of these data. Extended Data figures and tables are online-only (present in the online PDF and full-text HTML versions of the paper), peer-reviewed display items that provide essential background to the article but are not included in the main article due to space constraints. A maximum of ten Extended Data display items (figures and tables) is permitted.

Link Redacted

We hope to receive your revised manuscript within 4-8 weeks. If you cannot send it within this time, please let us know.

Nature Genetics is committed to improving transparency in authorship. As part of our efforts in this direction, we are now requesting that all authors identified as 'corresponding author' on published papers create and link their Open Researcher and Contributor Identifier (ORCID) with their account on the Manuscript Tracking System (MTS), prior to acceptance. ORCID helps the scientific community achieve unambiguous attribution of all scholarly contributions. You can create and link your ORCID from the home page of the MTS by clicking on 'Modify my Springer Nature account'. For more information, please visit www.springernature.com/orcid.

Sincerely,
Kyle

Kyle Vogan, PhD
Senior Editor
Nature Genetics
<https://orcid.org/0000-0001-9565-9665>

Referee expertise:

Referee #1: Genetics, complex traits, statistical methods

Referee #2: Genetics, complex traits, statistical methods

Referee #3: Genetics, complex traits, statistical methods

Reviewers' Comments:

Reviewer #1 (Remarks to the Author):

I appreciate the authors' detailed responses to my comments. The addition of independent replication data and other changes to the manuscript have strengthened it substantially.

My remaining comments are relatively minor:

Figure 3. I don't understand what the $h^2_{gwas,t}$ quantity represents or how it is calculated. Is this just the additive variance explained by the single genome-wide significant SNP? OK, reading further in the manuscript it seems like this is the single locus variance. Perhaps this can be made clear to the reader in the figure legend so the reader doesn't have to go searching for it?

The standard errors for e.g. the top leftmost SNP (cardiovascular) and top rightmost SNP (also cardiovascular) look odd. The SE is very small for $h^2_{gwas,t}$ for the top left SNP in comparison to the same quantity for the top rightmost SNP, yet the SE for $h^2_{gwas,t}$ is negligible for the top-left SNP but huge for the top right SNP. Is this correct? I'm confused as to how this can be the case?!

"For example, one class of works..." Rephrase?

Is it surprising that there doesn't seem to be a relationship between $h^2_{gxt,t}$ and $h^2_{gwas,t}$? What is $\rho = 0.02$ in this figure? Is this the correlation between the estimates? If so, this should be described in the figure legend.

Table 2. Gene names in italics please. "Effect allele" not "Effective allele"

What constitutes an "LD block" needs to be defined more precisely.

Imperfectly tagged causal SNPs: "We observed concordance of the estimates on the imputed dataset with those obtained on the array dataset (Supplementary Figure S10) with the two loci for SHBG showing significantly lower ME estimates." Why does the estimate of variance explained by the epistatic component decrease? This seems counterintuitive, e.g. when additional (imputed) SNPs are included in an additive GRM often the SNP heritability increases since more of the genome is tagged, but the opposite appears to be the case here. Why?

"We found that all five of our ME signals replicated in AoU with consistent direction of effects". I don't understand how there is a direction of effect in the epistatic part of the analysis? Also, I'd like to see p-values for all of these tests not just $p < \text{some threshold}$.

"The significant pairwise effects were almost always located on the same chromosome and close to the target SNP (Supplementary Table S7)." I think it is worth reporting these results in more detail.

Are the significant SNPs close by the index locus? This could point to the existence of e.g. long-range haplotype effects that are being picked up as "epistasis".

"rs72654473, downstream of the APOE gene on chromosome 9, shows a significant genetic interaction effect with rs6935921, near the LPA gene on chromosome 6." I think it would be worth graphing this interaction in a figure to see what its form is?

"We overlapped the 14 unique SNPs showing evidence of significant ME effects with sets of eQTLs and pQTLs". I'm not suggesting the following for this manuscript, but the authors should take advantage of the UKBB proteomic data to go looking for epistatic interactions there, particularly the ones identified in this manuscript as further evidence for their existence.

Reviewer #2 (Remarks to the Author):

The authors have demonstrated a commendable approach to addressing the previous review comments. I appreciate the thorough revisions, which substantially improve the manuscript's rigor and clarity:

1. Replication Datasets: The addition of both external and internal dataset replication analyses significantly strengthens the manuscript's methodological validation. This approach provides robust evidence for the proposed method's generalizability and reproducibility.

2. Simulation Scenarios: The expanded simulation scenarios offer a more comprehensive examination of the method's performance under various conditions, which enhances the manuscript's methodological credibility.

3. Clarity and Limitations: The authors have successfully addressed previous points of ambiguity and provided a transparent discussion of the method's limitations.

While potential avenues for future research exist — such as exploring long-range linkage disequilibrium, extending the calibration to binary traits, and investigating non-additive genetic encoding — these opportunities do not diminish the significant scientific contribution of the current work. On the contrary, they highlight the innovative nature of the current manuscript and provide a clear roadmap for future methodological developments. The manuscript represents a critical advancement in marginal epistasis analysis for large biobank-scale datasets, offering a high throughput approach that will undoubtedly stimulate further research and methodological innovations in the field.

Reviewer #2 (Remarks on code availability):

It would be useful to give examples of input and output datasets.

Reviewer #3 (Remarks to the Author):

None.

Version 4:

Decision Letter:

Our ref: NG-A63391R3

7th March 2025

Dear Sriram,

Thank you for submitting your revised manuscript "A biobank-scale test of marginal epistasis reveals genome-wide signals of polygenic epistasis" (NG-A63391R3). In light of your responses to Reviewer #1 and the accompanying clarifications and revisions, we will be happy in principle to publish your study in Nature Genetics as an Article pending final revisions to comply with our editorial and formatting guidelines.

We are now performing detailed checks on your paper, and we will send you a checklist detailing our editorial and formatting requirements soon. Please do not upload the final materials or make any revisions until you receive this additional information from us.

Thank you again for your interest in Nature Genetics. Please do not hesitate to contact me if you have any questions.

Sincerely,
Kyle

Kyle Vogan, PhD
Senior Editor
Nature Genetics
<https://orcid.org/0000-0001-9565-9665>

Reviewer #1:

Remarks to the Author:

-This is an interesting paper which purports to detect epistatic interactions for several traits within the UK Biobank

We thank the reviewer for their positive view of our work and for their thoughtful comments that have pushed us to substantially improve our manuscript.

Major comments:

-As far as I can see, the method the authors have employed is not particularly novel, basically taking an existing method and then restricting the search space to only those interactions involving a genome-wide significant SNP.

We apologize that our contribution was not clear in our previous submission. We emphasize that, to the best of our knowledge, FAME is the first algorithm capable of scalable marginal epistasis testing at UK Biobank-scale. This scalability allows us to analyze and validate signal robustness on large datasets, making our work the first to demonstrate convincing evidence of marginal epistasis effects across individual variants in human genetics.

Computationally, existing methods cannot be applied to the biobank-scale datasets that we examine even if their search spaces were restricted to GWAS-significant SNPs. In our previous submission, we showed that running the previous state-of-the-art method, MAPIT, would require 3 days to run on a small dataset (consisting of 20K individuals and 10K SNPs), making it impractical to run on whole-genome UKB genotype data. For Biobank-scale genotypes (N , $M \approx 500K$ SNPs), we extrapolate that MAPIT would take over 200 days and >200 GB RAM. On the other hand, FAME is able to perform these analyses in less than four hours with less than 32 GB of RAM. We now highlight this contribution in Section “Computational Efficiency”:

“We observe that the computational complexity of MAPIT grows rapidly with increasing sample size even for modest sample sizes and number of SNPs: requiring more than three days to run on 20K samples with 10K SNPs. This is, partly, because MAPIT did not provide the flexibility of modifying the variant testing strategy and, by default, tests the ME effect across all the SNPs provided. Thus, it is not feasible to run MAPIT on a large-scale dataset like UKBB (Supplementary Figure S8). Moreover, MAPIT requires loading the whole genotype matrix at once which we extrapolate would require more than 200 GB for UK Biobank size dataset.”

Methodologically, FAME builds on recent ideas from randomized linear algebra that we have used to efficiently fit mixed models in other contexts (Wu and Sankararaman, *Bioinformatics* 2018, Pazokitoroudi et al. *Nat. Comm.* 2020). Specifically, FAME uses a randomized method-of-moments estimator for biobank-scale estimation of marginal epistasis (ME). In our study, we made the choice to apply FAME to GWAS SNPs both as a demonstration of its capability and due to the ability to interpret results at these SNPs. Additionally, we have now performed extensive simulations to show that our method is well-calibrated (so that its false positive rate is controlled) even in settings where the model might be misspecified and have now included replication results that indicate that our results are “real” (see response to

comment below). While our focus in the current work has been on establishing the robustness of signals of ME, we expect that this approach can be extended to perform genome-wide scans of ME (analogous to a GWAS). We leave this direction for future work and have clarified our contributions in the revision—in the Discussion, we now write:

“Sixth, we have limited our analyses to GWAS significant SNPs in this study. The ability to estimate ME across the genome would allow us to obtain a more comprehensive nature of epistasis underlying complex traits.”

Biologically, our finding that the magnitude of ME is larger than GWAS effects (about 12x) on average is scientifically novel and substantially larger than what has been reported before.

-The authors lack a replication cohort, so it’s difficult to know whether the results are “real”. We thank the reviewer for pushing us to perform replication analysis which we agree is critical to establish the robustness of our results. In our revised manuscript, we have performed two replication experiments: an internal replication experiment (within the UKBB) and an external replication experiment in which we attempted to replicate our findings in UKBB in the All of Us data (for those phenotypes that had sufficiently large sample sizes for European ancestry individuals, $N > 50K$). In our internal replication experiment, we discovered three trait-locus pairs with genome-wide significant marginal epistasis (ME) signals in about 150K unrelated white British individuals in the UKBB. All signals replicated in a non-overlapping set of about 150K individuals with consistent direction of effects ($p < 0.05/3$) (see figure below). For our external replication, we selected five traits that had evidence of significant ME in UKBB for which we had a sufficiently large sample size of European ancestry individuals in AoU. We found that all five of our ME signals replicated in AoU with consistent direction of effects ($p < 0.05/5$). These replication experiments substantially strengthen our original results indicating that we are identifying individual SNPs that show strong and replicable signals of polygenic epistasis (see figure below). We describe these results in the Section titled “Replication of ME signals” and in Figure 4, which we reproduce here.

Figure 4: **Replicability of the findings from FAME.** We report estimates (and 95% CIs) of σ_{gxg}^2 for genome-wide significant ME loci ($p < \frac{5 \times 10^{-8}}{53}$) discovered in a discovery cohort and their estimates in a replication cohort. (Left). We use as discovery cohort about one half of the unrelated white British individuals in UKBB and as replication cohort a non-overlapping set of unrelated white British individuals in UKBB. b) We use as discovery cohort all unrelated white British individuals in UKBB and as replication cohort individuals of European ancestry in the All of Us (AoU) dataset.

- If we assume that the results are real, they lack biological insight above that provided by knowledge of the main effects. Would it be possible to follow up cases of interaction with 2x2 tests of epistasis to try and localize the interaction?

We thank the reviewer for this suggestion. In our revised manuscript, we performed pairwise epistasis analysis of the 16 genome-wide significant ME signals (GxGWAS) that we discovered using PLINK. Genome-wide significant pairwise effects were almost always located on the same chromosome and close to the target SNP (Supplementary Table S7 that we reproduce below). The only genome-wide significant genetic interactions involving a pair of SNPs that lie on distinct chromosomes is found in the case of SNP rs72654473 on lipoA. rs72654473, downstream of the APOE gene on chromosome 9, shows a significant genetic interaction effect with rs6935921, near the LPA gene on chromosome 6. We now report these results in the Section title “Localizing signals of ME”.

Trait	SNP ID	λ_{gc}	# of hits	
			Local	Distal
Alanine aminotransferase	rs3827385	1.05	0	0
Apolipoprotein B	rs964184	1.04	0	0
C-reactive protein	rs11208750	1.04	0	0
Cholesterol	rs964184	1.05	0	0
Hemoglobin A1c	rs34265667	1.08	0	0
Lipoprotein-A	rs628031	1.02	21	0
	rs146534110	1.24	23	0
	rs72654473	1.06	0	1
Mean platelet volume	rs463312	1.09	0	0
Monocyte count	rs79490353	1.11	0	0
SHBG	rs11656323	1.12	9	0
	rs35985803	1.06	5	0
Testosterone	rs11555142	1.06	0	0
	rs28990703	1.13	0	0
Triglycerides	rs964184	1.06	0	0
Urate	rs75246752	1.14	0	0

Table S7: **Pairwise interaction analysis.** We perform a GxGWAS where we test for association of the phenotype with pairs of SNPs across the genome (where one member of the pair is the target SNP showing significant ME for the phenotype). We report the genomic inflation factor (λ_{gc}) and the number of hits (after LD pruning). The number of hits is obtained from the interacting SNP pairs that are genome-wide significant ($p < 5 \times 10^{-8}/16$). The SNPs that interact with the target ME SNP are further pruned using the same parameters as the main analysis pipeline. Additionally, SNPs within the LD block of the target SNP and the MHC region were excluded from the final results.

To delve into the biological relevance of the interactive pairs, we conducted the following additional analyses:

1. Functional enrichment analysis:

- a. For the target ME SNPs, we overlapped our set of genome-wide significant ME signals with sets of eQTLs (cis and trans-eQTLs in whole blood identified in Vosa, Claringbould et al. *Nature Genetics* 2021) and pQTLs (cis and trans pQTLs for 57 plasma proteins identified in Yao et al. *Nat. Comm.* 2018 and 4,907 plasma proteins identified in Ferkingstad et al. *Nature Genetics* 2021). Eleven of the 14 ME SNPs were identified as regulating gene expression or protein levels. rs146534110, which is a ME SNP for Lipoprotein-A, is a cis-pQTL for LPA. rs964184, which is a ME SNP for three traits (Apolipoprotein B, Cholesterol, and Triglycerides), was found to be a cis-eQTL, a trans-eQTL, and a trans-pQTL. We describe these analyses in Section 2.7.4 titled “Functional interpretation of loci with significant ME effects”.

- b. For each GxGWAS result, we selected the top 1K SNPs that interact with the target SNP based on the interaction p-value (excluding SNPs from the LD block of the target SNP and the MHC region). We combined these sets of SNPs across all traits and used RegulomeDB (Boyle et al. *Genome Research* 2012) to test if the interacting SNPs were enriched for functional annotations at the SNP sites (measured as the fraction of sites with rank score $\geq 1f$) relative to a background of GWAS variants pooled across all outcomes (repeated 10,000 times). We identified significant functional enrichment with p-value=0.033. (We describe these analyses in Section 2.7.4 with more details).
 - c. To test whether the interactive SNPs are enriched for binding of specific transcription factors (TFs), we applied HOMER (Heinz et al. *Molecular cell* 2010) and analyzed a ± 100 bp window around the top 1,000 interactive SNPs for TF motif enrichments. We observed significant enrichment for multiple known (FDR<0.05) and de novo ($p < 1 \times 10^{-12}$) TF binding sites. Overall, these analyses revealed enrichments especially for TFs with helix-loop-helix domains as well as for GATA and nuclear hormone receptor TFs with zinc finger domains, which are known to be involved in the regulation of metabolic traits (Wang and Baker. *Developmental cell* 2015; Tong et al. *Science* 2000; Wei et al. *Cellular and molecular life sciences*, 2013).
2. Analyses of individual ME loci: We have now analyzed individual ME loci underlying Lipoprotein A, Lipids, Alanine aminotransferase, and Testosterone in the Section titled “Interpretation of individual ME loci”.

As one prominent example, rs3827385, which shows significant ME effect for alanine aminotransferase (ALT) and lies in the intron of SAMM50, a gene involved in mitochondrial structure, function, and fatty acid oxidation. This variant is 600 kb from and in moderate linkage disequilibrium ($r^2 = 0.36$, $D' = 0.67$) with a well-studied missense variant rs738409 (p.I148M) in the PNPLA3 gene, which has been linked to chronic liver disease (Romano et al. *Nature Genetics* 2008). Our analysis revealed significant ME for rs738409 on alanine aminotransferase ($p = 4.90 \times 10^{-24}$, $\sigma^2_{gxg} = 0.0148$).

Notably, prior work (Abul-Husn et al. *New England Journal of Medicine* 2018) has identified an interaction between rs738409 and rs72613567 (in HSD17B13) that affects liver function as measured by ALT and aspartate aminotransferase (AST) levels. Specifically, the rs72613567:TA allele attenuates the effect of rs738409 on aminotransferase levels and is linked with reduced PNPLA3 mRNA expression. A pairwise epistasis test in UKBB confirmed this interaction to be nominally significant ($\hat{\beta}_{pairwise} = -2.52 \times 10^{-2}$, $p = 1.94 \times 10^{-7}$; Supplementary Table S10 and Supplementary Figure S14 attached below).

To explore additional interactors, we investigated rs58542926, a missense variant in TM6SF2, previously identified to be associated with ALT and liver disease (Liu et al. *Nature Communications* 2014). The pairwise interaction between rs58542926 and

rs738409 on ALT was also nominally significant with rs58542926:T allele increasing the effect of rs738409 on ALT levels ($\hat{\beta}_{pairwise} = 2.91 \times 10^{-2}$, $p = 3.48 \times 10^{-4}$; Supplementary Table S10 in the revision and Supplementary Figure S14 attached below).

The proportion of phenotypic variance explained by the two interactive pairs is about 10% of the variance explained by ME at rs738409. Taken together, this example where individual pairwise interactions are nominally significant while the ME effect is genome-wide significant and the individual pairwise interactions explain only a modest fraction of the ME signal highlights the power of testing aggregated epistasis effects using FAME.

SNP_1	MAF_1	SNP_2	MAF_2	$\beta_{pairwise}$	$\sigma_{pairwise}^2$	$\sigma_{gg,t}^2$	$\sigma_{pairwise}^2/\sigma_{gg,t}^2$
rs72613567	0.28	rs738409	0.22	-2.52×10^{-2}	0.09×10^{-3}	1.48×10^{-3}	0.059
rs58542926	0.08	rs738409	0.22	2.91×10^{-2}	0.04×10^{-3}	1.48×10^{-3}	0.029

Table S10: **Summary of pairwise interaction analysis for Alanine Aminotransferase.** We report the statistics for the pairwise interaction effects on alanine aminotransferase of rs738409 and each of two SNPs, rs72613567 and rs58542926. The pairwise variance explained, $\sigma_{pairwise}^2$, is calculated as $4MAF_1(1-MAF_1)MAF_2(1-MAF_2)\beta_{pairwise}^2$, where the SNP pairs are independent ($LD \approx 0$).

Figure S14: **Pairwise epistasis test.** We analyze the relationship between (a) rs72613567 and rs738409 and (b) rs58542926 and rs738409 on alanine aminotransferase. The error bar represents the 95% confidence interval from 1,000 bootstrap samples.

-Why not evaluate the significance of the tests using permutation i.e. permuting first the phenotype with respect to the genetic data, and then permuting the phenotype and genome-wide significant variant with respect to the rest of the genotype data?

Performing permutation tests on biobank-scale when testing each trait-SNP combination is computationally demanding. However, we can use permutations to demonstrate the robustness of the detected findings. To address the reviewer's suggestion, we permuted the target variant to construct the permuted ME matrix, while keeping both the additive matrix (including the target variant) and phenotype unchanged. This allowed us to jointly estimate the additive and permuted ME components. Notably, none of the significant trait-locus pairs remained significant after these permutations ($p < 0.05/16$; Supplementary Table S5 reproduced below). This analysis provides additional evidence of the validity of our method and directly addresses the reviewer's request to permute the epistatic component while preserving the additive component. We have now included the results of this analysis in the revision in the Section titled "Permutation tests to assess the estimates of ME".

Trait	SNP ID	p-value
Alanine aminotransferase	rs3827385	0.30
Apolipoprotein B	rs964184	0.75
C-reactive protein	rs11208750	0.60
Cholesterol	rs964184	0.64
Hemoglobin A1c	rs34265667	0.60
Lipoprotein-A	rs628031	0.18
	rs146534110	0.59
	rs72654473	0.14
Mean platelet volume	rs463312	0.07
Monocyte count	rs79490353	0.27
SHBG	rs11656323	0.28
	rs35985803	0.36
Testosterone	rs11555142	0.63
	rs28990703	0.03
Triglycerides	rs964184	0.19
Urate	rs75246752	0.99

Table S5: **Permutation test of ME signals.** We permute the marginal epistasis (ME) matrix while keeping the rest of the components the same. We then run FAME on the 16 significant loci-pairs report earlier. None of the pairs are significant after permutation ($p < \frac{0.05}{16}$).

Minor comments:

Introduction:

As far as I'm aware, the first studies to look at the feasibility of performing an exhaustive pairwise search for epistasis in GWAS data are Marchini et al. 2005 Nat Genet and Evans et al. 2006 PLoS Genet. These should both be cited.

Thank you. We have now cited these papers in the introduction.

Results:

-“...are the residual variance, genetic variance and the ME variance”. Perhaps specify “additive genetic variance” to make this explicit for the reader.

We explicitly state “additive genetic variance”, as the reviewer suggested.

-Make clear the matrix symbol is the Hadamard product

Thank you, we have remedied this.

-Is there some inflation of the test statistic in the bottom row of Figure S2?

We appreciate the reviewer’s careful scrutiny. In our updated pipeline, for each target variant tested for interaction effects, we further regressed out the additive effect from the LD block where the variant is located. While our previous approach only excluded the LD block from the analysis of interaction effects. This update effectively addresses the slight inflation issue noted by the reviewer, as can be seen from our updated Figure S2 (reproduced below).

Figure S2: **Calibration of p-values for the test of ME based on inclusion of the LD block surrounding the target SNP.** In this experiment, we compared the results of applying FAME in two different scenarios. In scenario one, we constructed the ME matrix by multiplying the genotypes at each target SNP with the remaining SNPs in the genome and estimated the ME variance component and corresponding p-values (top row). In scenario two, the ME matrix is constructed by multiplying the genotype at each target SNP with all other SNPs except those within the LD block of the target SNP, with main effects of the SNPs within the LD block regressed out. This is the standard pipeline we applied to analyze real traits (bottom row). In both experiments, the additive component (σ_g^2) for all SNPs is fit jointly with the ME component $\sigma_{g \times g, t}^2$. The simulated phenotype has additive but no genetic interactions. The target SNPs were chosen from the significant LD-pruned SNPs in a GWAS.

We discuss these results in Section 2.2 as:

“While FAME is calibrated when the target ME SNPs were selected at random (Supplementary Figure S1), the p-values tend to be inflated when the target SNPs were selected based on a GWAS (Section 2.6, Supplementary Figure S2). To address this issue, we regressed out additive effects at SNPs that lie within the LD block around the target SNP and excluded these SNPs when constructing the set of genetic interactions E_t . This approach effectively controlled the false positive rate with no significant ME signal detected across any of the null simulation settings (Figure 2a).”

-The final paragraph of section 2.2. My memory of this is that the inflation is only an issue when one of the contributing SNPs has a massive effect. Did the authors' simulate a large main effect when evaluating the method's robustness to this?

We have now performed simulations under this setting of large additive main effects (in addition to several other types of model misspecification, including non-linear covariates that impact the phenotype, GxE with a hidden E variable, heavy-tailed environmental noise, and heteroskedastic environmental noise that we describe in the Section “Robustness to model misspecification” in the main text). We simulated large additive genetic effects and further considered large additive effects that are imperfectly tagged. Finally, we also considered a setting where the dependence of additive effect sizes on MAF and LD is different (LDAK model) than the one assumed by FAME. We find that FAME remains calibrated in each of these settings. We reproduce the QQ plots below and show additional metrics relevant to type-I error rate and λ_{gc} in the Figure S4 and Table S1 below (the last three rows of the Table are simulations with large additive effects while the remaining rows show other types of model misspecification that we explored). More details on these simulations are included in Supplementary Text S1.

Figure S4: **Robustness to mis-specification.** We explored the impact of three types of model mis-specifications (Large additive effects, Imperfectly-tagged large additive effects, and Imperfectly-tagged LDAK model) on the calibration of FAME.

α	0.5	0.1	0.05	0.01	0.005	λ_{gc}	CI(λ_{gc})
Polynomial covariate relationship	1.05	1.02	0.88	1.10	0.55	1.11	(0.874, 1.456)
G-E with unobserved E	0.97	0.45	0.69	0.69	0.69	0.95	(0.669, 1.287)
Heavy-tailed noise	0.93	0.71	0.68	0.62	0.93	0.82	(0.607, 1.002)
Heteroskedastic noise	0.98	1.08	1.29	0.54	0.54	0.96	(0.740, 1.232)
Large additive effects	1.02	0.89	0.77	0.77	0.77	1.03	(0.855, 1.194)
Imperfectly-tagged large additive effects	0.97	0.82	0.66	0.73	0.73	0.92	(0.766, 1.144)
Imperfectly-tagged LDAK model	1.02	0.79	0.98	0.75	0.0	1.05	(0.881, 1.439)

Table S1: **Robustness of FAME under model misspecification:** We test the robustness of FAME under seven types of model misspecification. We report the ratio of observed to the expected type-I error rate at different p-value thresholds ($\alpha \in \{0.5, 0.1, 0.05, 0.01, 0.005\}$) and the genomic inflation factor (λ_{gc}). Across all types of model misspecification, the genomic inflation factor is close to 1 (the 95% CI of the inflation factor overlaps 1 across each of the settings). Similarly, the ratio of observed type-I error rate is close to the expected type-I error rate (none of the observed type-I error rate is significantly larger than the expectation across all thresholds and model misspecifications ($p < 0.05$)).

-Section 2.3: Specify what the sample size is.

We clarify that the sample size considered here is the set of unrelated white British individuals ($N=291,273$). We have now added the dataset being used in the “Power analysis” section with a detailed description in the “Datasets” section of Materials and Methods.

-Section 2.4: Can the authors give some indication of the memory requirements?

FAME requires <32 GB of memory for analyzing all the datasets that we have considered (both $\approx 300K$ unrelated white British individuals genotyped at $\approx 500K$ array SNPs and on $\approx 5M$ imputed SNPs). We have specified the memory requirements in Section 2.5 “Computational efficiency”.

-Section 2.5.5 “has been association”

Thank you. We have made this correction.

Methods:

-Could the authors please list the 53 traits analyzed in a supplementary table? (apologies if I have missed this). Why were these particular traits chosen?

We have now added the table containing the traits analyzed to the supplementary information (reproduced below). These traits were chosen because they have been analyzed by us and others to estimate aspects of genetic architecture that include additive (Sinnott-Armstrong et al. *Nature Genetics* 2021, Wei et al. *eLife* 2023), dominance (Pazokitoroudi et al. *AJHG* 2021), and gene-environment interactions (Pazokitoroudi et al. *AJHG* 2024) and span a diverse set of phenotypic categories. We now describe this in the Section “Datasets” in Materials and Methods of the Main Text.

Trait	Category	Trait	Category
BMD Heel T-score	Anthropometry	LDL direct	Cardiovascular
Basal metabolic rate	Anthropometry	Lipoprotein-A	Cardiovascular
Body mass index	Anthropometry	Systolic blood pressure	Cardiovascular
Height	Anthropometry	Triglycerides	Cardiovascular
Eosinophil count	Blood biochemistry	Glucose	Diabetes
High light scatter reticulocyte count	Blood biochemistry	Hemoglobin A1c	Diabetes
IGF-1	Blood biochemistry	Corneal Hysteresis	Eye
Lymphocyte count	Blood biochemistry	Alanine aminotransferase	Liver
Mean corpuscular hemoglobin	Blood biochemistry	Albumin	Liver
Mean platelet volume	Blood biochemistry	Aspartate aminotransferase	Liver
Mean sphered cell volume	Blood biochemistry	Direct bilirubin	Liver
Monocyte count	Blood biochemistry	Total bilirubin	Liver
Platelet count	Blood biochemistry	Creatinine	Renal
Platelet distribution width	Blood biochemistry	Creatinine in urine	Renal
RBC count	Blood biochemistry	Cystatin-C	Renal
RBC distribution width	Blood biochemistry	Microalbumin in urine	Renal
SHBG	Blood biochemistry	Phosphate	Renal
Testosterone	Blood biochemistry	Potassium in urine	Renal
White blood cell count	Blood biochemistry	Sodium in urine	Renal
Alkaline phosphatase	Bone	Total protein	Renal
Calcium	Bone	Urate	Renal
Apolipoprotein A	Cardiovascular	Urea	Renal
Apolipoprotein B	Cardiovascular	Age first birth	Other
C-reactive protein	Cardiovascular	FEV1-FVC ratio	Other
Cholesterol	Cardiovascular	FVC	Other
Diastolic blood pressure	Cardiovascular	Tanning (quantitative)	Other
HDL cholesterol	Cardiovascular		

Table S9: Traits analyzed in this study.

Figure 2b - what is the sample size? Annotate this in the legend?

The sample size is $N=291,273$ corresponding to unrelated white British individuals in UKBB. We have added this information to the caption.

Table 1 - Please insert the marginal results in this table including the MAF of the SNP, its (fixed) effect size and p value. Are these known hits? What are the closest genes? Also please list the rs numbers of that variants, not just the base pair position (which build?). Similar comments for Table 2.

We now incorporated these additional pieces of information into the main tables, as the reviewer suggested. Specifically, in all the tables, we reported rsID by default. We included the MAF of the SNP in Table 1 (reproduced below). Additional information, e.g., fixed effect size and standard error, the closest genes, and variant type in Table 2 (reproduced below).

We did additional validation using the GWAS Catalog and the Open Targets Genetics Platform, as well as external GWAS summary statistics (Sinnott-Armstrong et al. *Nature Genetics* 2021). We confirmed all the candidate trait-loci pairs are known hits (either listed in the GWAS catalog or in Open Targets or found to be significant from external GWAS summary statistics).

Table 1: **Candidate trait-loci pairs with significant marginal epistasis (ME)**. The 16 GWAS candidate trait-loci pairs passing the significant threshold $\frac{5 \times 10^{-8}}{53}$ are displayed in this table, along with the minor allele frequency (MAF) for each SNP. For each pair, we reported the heritability explained by ME effect across each pair as $h_{g_{gg,t}}^2$, and the corresponding p-value as $P_{g_{gg}}$. In addition, we compared the corresponding marginal additive heritability $h_{g_{was,t}}^2$ and the corresponding p-value $P_{g_{was}}$. Finally, we presented the ratio $\frac{h_{g_{gg,t}}^2}{h_{g_{was,t}}^2}$, labeled as "Ratio".

Trait	SNP ID	MAF	$h_{g_{gg,t}}^2$ $\times 0.001$	$P_{g_{gg}}$	$h_{g_{was,t}}^2$ $\times 0.001$	$P_{g_{was}}$	Ratio
Alanine aminotransferase	rs3827385	0.18	10.01	1.35×10^{-13}	1.17	9.30×10^{-82}	8.57
Apolipoprotein B	rs964184	0.13	7.43	9.05×10^{-10}	1.45	1.82×10^{-89}	5.12
C-reactive protein	rs11208750	0.20	8.70	3.66×10^{-10}	1.16	5.17×10^{-73}	7.52
Cholesterol	rs964184	0.13	9.01	2.61×10^{-13}	1.15	7.50×10^{-74}	7.84
Hemoglobin A1c	rs34265667	0.03	5.09	5.52×10^{-12}	1.40	2.58×10^{-94}	3.64
Lipoprotein-A	rs628031	0.41	13.80	1.50×10^{-12}	0.31	8.07×10^{-17}	43.94
	rs146534110	0.01	6.65	6.50×10^{-26}	5.33	3.77×10^{-259}	1.25
	rs72654473	0.11	9.45	5.01×10^{-12}	1.31	3.49×10^{-65}	7.18
Mean platelet volume	rs463312	0.05	6.83	2.66×10^{-15}	1.42	1.47×10^{-89}	4.81
Monocyte count	rs79490353	0.03	3.85	8.44×10^{-10}	2.77	1.79×10^{-183}	1.39
SHBG	rs11656323	0.05	8.22	1.73×10^{-17}	0.41	4.30×10^{-29}	20.02
	rs35985803	0.08	7.04	3.43×10^{-10}	1.01	8.92×10^{-69}	7.00
Testosterone	rs11555142	0.09	7.72	1.73×10^{-11}	0.18	1.62×10^{-29}	41.92
	rs28990703	0.04	8.64	1.03×10^{-23}	0.26	6.11×10^{-41}	33.28
Triglycerides	rs964184	0.13	9.11	2.58×10^{-13}	15.33	$\leq 1 \times 10^{-259}$	0.59
Urate	rs75246752	0.01	3.20	1.62×10^{-10}	0.72	1.28×10^{-62}	4.44

Table 2: **Information on significant trait-locus pairs.** For each significant trait-locus pair, we provide the GWAS allelic effect size estimate and corresponding standard error (SE), the closest protein-coding gene to the SNP, and the variant type. The effective allele of the variant is displayed in column "Allele", for effect direction reference.

Trait	SNPID	Allele	Chr:Pos (b37)	$\hat{\beta} \pm SE$ $\times 0.001$	Nearest gene	Variant type
Alanine aminotransferase	rs3827385	T	22:44388817	62.70 \pm 3.27	SAMM50	Intron
Apolipoprotein B	rs964184	C	11:116648917	79.44 \pm 3.96	ZPR1	3' UTR
C-reactive protein	rs11208750	C	1:66257838	-60.66 \pm 3.36	PDE4B	Regulatory
Cholesterol	rs964184	C	11:116648917	70.62 \pm 3.88	ZPR1	3' UTR
Hemoglobin A1c	rs34265667	G	8:41542093	-145.27 \pm 7.05	ANK1	Synonymous
Lipoprotein-A	rs628031	G	6:160560845	25.38 \pm 3.05	SLC22A1	Missense
	rs146534110	G	6:160578069	468.07 \pm 13.59	SLC22A1	Intron
	rs72654473	C	19:45414399	-83.11 \pm 4.87	APOE	Non-coding exon
Mean platelet volume	rs463312	A	20:57597970	-124.95 \pm 6.22	TUBB1	Missense
Monocyte count	rs79490353	T	13:28623048	236.49 \pm 8.18	FLT3	Intron
SHBG	rs11656323	T	17:7145117	-66.54 \pm 5.94	GABARAP	5' UTR
	rs35985803	G	17:7254315	-81.97 \pm 4.68	ACAP1	Missense
Testosterone	rs28990703	A	12:2977954	61.76 \pm 4.61	FOXM1	Intron
	rs11555142	G	7:99032593	-34.35 \pm 3.04	ATP5J2-PTCD1	Synonymous
Triglycerides	rs964184	C	11:116648917	257.75 \pm 3.81	ZPR1	3' UTR
Urate	rs75246752	G	1:145630111	-167.92 \pm 10.05	RNF115	Intron

Table S2: g count- are these independent GWS SNPs? Please indicate

Yes, we apologize for the unclear statement. We have now indicated they are independent SNPs in the caption, and have moved it as table S4.

Table S3: Same comments as above with respect to listing the rs numbers, closest genes, etc. We now add this information in Table 2, as displayed above. We have now moved the old Table S3 as Table S6.

Limitations:

-Presumably the method will not detect interactions that do not produce large main effects. Perhaps this should be commented on.

We want to clarify that using GWAS significant loci only serves to screen for a set of target SNPs to be used for testing for ME (we could envision testing for ME at other SNPs in the genome based on prior biological knowledge). The model underlying FAME does not require a large main effect. To validate this argument, we used the same simulations that were employed for our power analysis. In these simulations, additive effects are chosen independently of the target ME SNP. We conducted a marginal association analysis on the target variants. Among the significant ME loci (p-value < 5e-08), 32.5% were also significant in main effect, while 27.6% had main effect p-values > 0.05. This analysis demonstrates that FAME can detect interactions even in the absence of strong main effects.

-Is the method limited to quantitative traits? Again, perhaps commented on.

While our method can be applied to binary traits, we need to explore its calibration when the trait prevalence is low as well as the impact of ascertainment. As a result, our current study focuses on quantitative traits. We have emphasized this in the abstract and comment on this limitation in the Discussion, where we write:

“...Third, we have only applied FAME to quantitative traits which match the assumptions underpinning our model. Assessing the applicability of this model to binary (disease) traits would require exploring the impact of trait prevalence and ascertainment...”

Reviewer #2:

Remarks to the Author:

The paper presents a study on marginal epistasis (ME) at a biobank scale, revealing significant genome-wide signals of polygenic epistasis. The authors have utilized a large dataset from the UK Biobank and developed a method to efficiently estimate the covariance matrix for ME. They have analyzed the heritability at loci with significant ME effects, comparing these with heritability estimates based on GWAS effects. Overall, the paper presents significant findings in the field of genetic epidemiology, particularly in understanding polygenic epistasis. However, addressing the below points would greatly enhance the study's clarity, robustness, and impact.

We thank the reviewer for their encouraging comments.

1. The authors exclude SNPs within the LD block around the target SNP when constructing the set of genetic interactions but retain these SNPs in the additive component. This approach might overlook epistatic interactions in long-range linkage disequilibrium (LD) regions (Singhal et al, AJHG 2023).

Thank you for bringing this up. In our updated pipeline, for each target variant tested for interaction effects, we further regressed out the additive effect from the LD block where the variant is located. We show that our approach of excluding SNPs in the LD block and regressing out additive effects in the LD block is important to ensure calibration (controlling the false positive rate when there is no epistasis; see Figure in response to comment #2 below). Further, in our revised manuscript, we have undertaken extensive simulations to show that our method remains calibrated under a wide range of model mis-specifications (see Supplementary Table S1 and Section 2.3 in the revised manuscript).

We agree that our approach is likely to miss epistatic interactions of the target SNP with other SNPs within the LD block but would be sensitive to interactions of the target SNP to SNPs outside the LD block. This approach might lead to lower power to detect epistatic interactions in long-range LD regions if the interacting SNPs fall within the LD block (e.g. the 250 kb threshold employed by Singhal et al. 2023). We view this as an issue of reduced power and not of inflation so that our method will be sensitive to specific types of epistasis (epistasis that involves interactions that span the LD block). We view the goal of finding robust signals of ME as an important first step. It would be of high interest to explore whether we can use SNP pairs that show long-range LD to form our ME matrix to further boost the power to detect epistasis. We discuss this limitation in Discussion session, writing:

“First, our approach of regressing out additive effects in the LD block surrounding the target SNP and only testing for interactions outside the LD block is important for calibration but might reduce power to detect specific signals of epistasis, e.g., epistatic signals that are local to an LD block or epistatic signals that lie in regions of long-range linkage disequilibrium”

2. Supplementary Figure S2 indicates residual inflation in the Causal Ratio 0.1 scenario. The paper should address how this inflation is corrected or managed, ensuring the robustness of the findings.

We appreciate the reviewer's careful scrutiny. In our updated pipeline, for each target variant tested for interaction effects, we further regressed out the additive effect from the LD block

where the variant is located. While our previous approach only excluded the LD block from the analysis of interaction effects. This update effectively addresses the slight inflation issue noted by the reviewer, presented in the Causal Ratio 0.1 Scenario, as can be seen from our updated Figure S2 (reproduced below).

Figure S2: **Calibration of p-values for the test of ME based on inclusion of the LD block surrounding the target SNP.** In this experiment, we compared the results of applying FAME in two different scenarios. In scenario one, we constructed the ME matrix by multiplying the genotypes at each target SNP with the remaining SNPs in the genome and estimated the ME variance component and corresponding p-values (top row). In scenario two, the ME matrix is constructed by multiplying the genotype at each target SNP with all other SNPs except those within the LD block of the target SNP, with main effects of the SNPs within the LD block regressed out. This is the standard pipeline we applied to analyze real traits (bottom row). In both experiments, the additive component (σ_g^2) for all SNPs is fit jointly with the ME component $\sigma_{g \times g, t}^2$. The simulated phenotype has additive but no genetic interactions. The target SNPs were chosen from the significant LD-pruned SNPs in a GWAS.

We discuss this in Section 2.2 Calibration of FAME, writing:

“While FAME is calibrated when the target ME SNPs were selected at random (Supplementary Figure S1), the p-values tend to be inflated when the target SNPs were selected based on a GWAS (Section 2.6), Supplementary Figure S2).

To address this issue, we regressed out additive effects at SNPs that lie within the LD block around the target SNP and excluded these SNPs when constructing the set of genetic interactions E_t . This approach effectively controlled the false positive rate with no significant ME signal detected across any of the null simulation settings (Figure 2a).”

3. Based on Supplementary Figure S5, the method appears underpowered for rare variants. The paper should discuss whether this underpowered nature extends to the detection of recessive effects of variants, especially when SNPs are coded for non-additive effects.

We appreciate the reviewer’s concern regarding the possibly lower power observed in certain rare variant settings. Firstly, Supplementary Figure S5 in our previous manuscript (Supplementary Figure S11 in the revision) focuses on calibration i.e. on whether false positive rate of our method is controlled in a setting where there is no epistasis and the additive genetic effects are limited to low-frequency variants. We acknowledge that the calibration of our method for ultra rare variants remains underexplored, as our focus has been on variants with a minor allele frequency (MAF) of 0.01 or higher.

That said, as shown in our updated Table 1 (reproduced below), several of the variants in our analysis fall within the lower MAF range, with two notable trait-variant pairs (Lipoprotein-A: rs146534110, Urate: rs75246752) having MAF around 0.01. However, it is also possible that ME is more likely when the loci are rare.

Table 1: Candidate trait-loci pairs with significant marginal epistasis (ME). The 16 GWAS candidate trait-loci pairs passing the significant threshold $\frac{5 \times 10^{-8}}{53}$ are displayed in this table, along with the minor allele frequency (MAF) for each SNP. For each pair, we reported the heritability explained by ME effect across each pair as $h_{g \times g, t}^2$, and the corresponding p-value as $P_{g \times g}$. In addition, we compared the corresponding marginal additive heritability $h_{gwas, t}^2$ and the corresponding p-value P_{gwas} . Finally, we presented the ratio $\frac{h_{g \times g, t}^2}{h_{gwas, t}^2}$, labeled as "Ratio".

Trait	SNP ID	MAF	$h_{g \times g, t}^2$ $\times 0.001$	$P_{g \times g}$	$h_{gwas, t}^2$ $\times 0.001$	P_{gwas}	Ratio
Alanine aminotransferase	rs3827385	0.18	10.01	1.35×10^{-13}	1.17	9.30×10^{-82}	8.57
Apolipoprotein B	rs964184	0.13	7.43	9.05×10^{-10}	1.45	1.82×10^{-89}	5.12
C-reactive protein	rs11208750	0.20	8.70	3.66×10^{-10}	1.16	5.17×10^{-73}	7.52
Cholesterol	rs964184	0.13	9.01	2.61×10^{-13}	1.15	7.50×10^{-74}	7.84
Hemoglobin A1c	rs34265667	0.03	5.09	5.52×10^{-12}	1.40	2.58×10^{-94}	3.64
Lipoprotein-A	rs628031	0.41	13.80	1.50×10^{-12}	0.31	8.07×10^{-17}	43.94
	rs146534110	0.01	6.65	6.50×10^{-26}	5.33	3.77×10^{-259}	1.25
	rs72654473	0.11	9.45	5.01×10^{-12}	1.31	3.49×10^{-65}	7.18
Mean platelet volume	rs463312	0.05	6.83	2.66×10^{-15}	1.42	1.47×10^{-89}	4.81
Monocyte count	rs79490353	0.03	3.85	8.44×10^{-10}	2.77	1.79×10^{-183}	1.39
SHBG	rs11656323	0.05	8.22	1.73×10^{-17}	0.41	4.30×10^{-29}	20.02
	rs35985803	0.08	7.04	3.43×10^{-10}	1.01	8.92×10^{-69}	7.00
Testosterone	rs11555142	0.09	7.72	1.73×10^{-11}	0.18	1.62×10^{-29}	41.92
	rs28990703	0.04	8.64	1.03×10^{-23}	0.26	6.11×10^{-41}	33.28
Triglycerides	rs964184	0.13	9.11	2.58×10^{-13}	15.33	$\leq 1 \times 10^{-259}$	0.59
Urate	rs75246752	0.01	3.20	1.62×10^{-10}	0.72	1.28×10^{-62}	4.44

The final point that the reviewer raises on the impact of encoding of SNPs for non-additive (e.g. recessive effects) is very interesting and more generally, whether such encodings would allow us to detect interactions that are not apparent from additive encodings.

We view the investigation of epistasis involving rare variants and the impact of coding as important directions of future work. We have now explicitly acknowledged the limited exploration of FAME in the context of rare variant epistasis and alternate genotype codings in the discussion section of the revised manuscript, writing:

“Fourth, while we have focused on testing for ME at common SNPs, it would be of great interest to apply FAME to test for ME at rare variants and to explore interactions with alternative genotypic encodings (interactions with recessive or dominance effects).”

4. The heritability estimates from gene-by-gene interactions (g_{xg}) are reported to be 12 times larger than those from GWAS, despite using GWAS significant SNPs. The paper should justify why this significant difference is not a result of false positives.

In our revised manuscript, we now perform additional simulations (under a scenario where there are additive effects but no marginal epistasis) that demonstrate that the ratio of the heritability estimates from g_{xg} and GWAS are not inflated. We reproduce the key figure and table below and reference these results in Section 2.3 Robustness to model misspecification as:

“Since it is of interest to compare the strength of ME to the marginal GWAS, we also evaluated the estimates of the ratio $\sigma_{g_{xg}}^2 / \sigma_{g_{was}}^2$ under the regular additive model (Supplementary Figure S7 and Supplementary Table S3) and under each of the model misspecifications considered above (Supplementary Figure S7 and Supplementary Table S3) to find that FAME produces calibrated p-values.”QUOTE

Figure S7: **Calibration of ratio** $\frac{\hat{\sigma}_{g_{xg}}^2}{\hat{\sigma}_{g_{was}}^2}$. We examined the calibration of estimates of the ratio: $\frac{\hat{\sigma}_{g_{xg}}^2}{\hat{\sigma}_{g_{was}}^2}$ (a) under simulations where the model is correctly specified with causal ratio 0.1 and heritability 0.05 on *UKBB-small* data; and (b) simulations that consider model misspecifications.

α	0.5	0.1	0.05	0.01	0.005	0.001	λ_{gc}	CI(λ_{gc})
Correctly specified model	0.94	0.95	0.86	0.60	0.67	0.67	0.88	(0.781, 0.982)
Mis-specified models	0.98	0.82	0.80	0.76	0.70	1.17	0.96	(0.884, 1.033)

Table S3: **Calibration of tests of the hypothesis that $\frac{\sigma_{gwg}^2}{\sigma_{gwas}^2} = 0$.** We test the calibration of p-values of tests of the hypothesis that $\frac{\sigma_{gwg}^2}{\sigma_{gwas}^2} = 0$. We report the ratio of observed to the expected type-I error rate at different p-value thresholds ($\alpha \in \{0.5, 0.1, 0.05, 0.01, 0.005\}$). In the first row, we report these statistics under a correctly specified null model (causal ratio=0.1, $h^2 = 0.05$ with no ME); in the second row, we report the combined statistics across all settings where the model is mis-specified. Across all types of model misspecification, the genomic inflation factor is close to 1 (the 95% CI of the inflation factor overlaps 1 across each of the settings) while none of the observed type-I error rates is significantly larger than the expectation across all thresholds ($p < 0.05$).

5. The authors claim that the epistatic signal detected is likely polygenic and that their approach of testing aggregate effects is more powerful. However, this is tested only with GWAS significant variants. The paper should explore whether the same pattern is observed with non-significant variants to strengthen the argument.

To further justify our claim, we additionally conducted GxGWAS for each of the ME significant trait-loci pairs. Genome-wide significant pairwise effects were almost always located on the same chromosome and close to the target SNP (Supplementary Table S7 that we reproduce below). The only genome-wide significant genetic interactions involving a pair of SNPs that lie on distinct chromosomes is found in the case of SNP rs72654473 on lipoA. rs72654473, downstream of the APOE gene on chromosome 9, shows a significant genetic interaction effect with rs6935921, near the LPA gene on chromosome 6. We now report these results in the Section title “Localizing signals of ME”.

Trait	SNP ID	λ_{gc}	# of hits	
			Local	Distal
Alanine aminotransferase	rs3827385	1.05	0	0
Apolipoprotein B	rs964184	1.04	0	0
C-reactive protein	rs11208750	1.04	0	0
Cholesterol	rs964184	1.05	0	0
Hemoglobin A1c	rs34265667	1.08	0	0
Lipoprotein-A	rs628031	1.02	21	0
	rs146534110	1.24	23	0
	rs72654473	1.06	0	1
Mean platelet volume	rs463312	1.09	0	0
Monocyte count	rs79490353	1.11	0	0
SHBG	rs11656323	1.12	9	0
	rs35985803	1.06	5	0
Testosterone	rs11555142	1.06	0	0
	rs28990703	1.13	0	0
Triglycerides	rs964184	1.06	0	0
Urate	rs75246752	1.14	0	0

Table S7: **Pairwise interaction analysis.** We perform a GxGWAS where we test for association of the phenotype with pairs of SNPs across the genome (where one member of the pair is the target SNP showing significant ME for the phenotype). We report the genomic inflation factor (λ_{gc}) and the number of hits (after LD pruning). The number of hits is obtained from the interacting SNP pairs that are genome-wide significant ($p < 5 \times 10^{-8}/16$). The SNPs that interact with the target ME SNP are further pruned using the same parameters as the main analysis pipeline. Additionally, SNPs within the LD block of the target SNP and the MHC region were excluded from the final results.

Another line of evidence for the polygenicity of epistasis arises from our analyses of individual ME loci. Here we highlight the example of rs3827385, which shows significant ME effect for alanine aminotransferase and lies in the intron of SAMM50, a gene involved in mitochondrial structure, function, and fatty acid oxidation. This variant is 600 kb from and in moderate linkage disequilibrium ($r^2 = 0.36$, $D' = 0.67$) with the missense variant rs738409 (p.I148M) in the PNPLA3 gene, which has been linked to chronic liver disease. Our analysis revealed significant ME for rs738409 on alanine aminotransferase ($p = 4.90 \times 10^{-24}$, $\sigma_{gxg}^2 = 0.0148$). Notably, prior work (Abul-Husn et al. *New England Journal of Medicine* 2018) has identified an interaction between rs738409 and rs72613567 (in HSD17B13) that affects liver function as measured by ALT and aspartate aminotransferase (AST) levels. Specifically, the rs72613567:TA allele attenuates the effect of rs738409 on aminotransferase levels and is linked with reduced PNPLA3 mRNA expression. A pairwise epistasis test in UKBB confirmed this interaction to be nominally significant ($\hat{\beta}_{pairwise} = -2.52 \times 10^{-2}$, $p = 1.94 \times 10^{-7}$; Supplementary Table S10 in the revision and Supplementary Figure S14 attached below). To explore additional interactors, we investigated rs58542926, a missense variant in TM6SF2, previously identified to be associated with ALT and liver disease (Liu et al. *Nature Communications* 2014). The pairwise interaction between rs58542926 and rs738409 on ALT was also nominally significant with rs58542926:T allele increasing the effect of rs738409 on ALT levels ($\hat{\beta}_{pairwise} = 2.91 \times 10^{-2}$, $p = 3.48 \times 10^{-4}$; Supplementary Table S10 in the revision and Supplementary Figure S14 attached below). The proportion of phenotypic variance explained by the two interactive pairs is about 10% of the variance explained by ME at rs738409. Taken together, this example where individual pairwise interactions are nominally significant while the ME effect is genome-wide significant and the individual pairwise interactions explain only a modest fraction of the ME signal highlights the power of testing aggregated epistasis effects using FAME.

Regarding the question of whether the same pattern would be observed for non-significant variants, we note that this may also depend on the distinct epistatic architecture present in GWAS-significant versus non-significant variants. However, we acknowledge that further exploration could reveal interesting differences in epistatic patterns between these variant groups. We now write it in the discussion section as: “Sixth, we have limited our analyses to GWAS significant SNPs in this study. The ability to estimate ME across the genome would allow us to obtain a more comprehensive understanding of the nature of epistasis underlying complex traits.”

SNP_1	MAF_1	SNP_2	MAF_2	$\beta_{pairwise}$	$\sigma_{pairwise}^2$	$\sigma_{gxg,t}^2$	$\sigma_{pairwise}^2/\sigma_{gxg,t}^2$
rs72613567	0.28	rs738409	0.22	-2.52×10^{-2}	0.09×10^{-3}	1.48×10^{-3}	0.059
rs58542926	0.08	rs738409	0.22	2.91×10^{-2}	0.04×10^{-3}	1.48×10^{-3}	0.029

Table S10: **Summary of pairwise interaction analysis for Alanine Aminotransferase.** We report the statistics for the pairwise interaction effects on alanine aminotransferase of rs738409 and each of two SNPs, rs72613567 and rs58542926. The pairwise variance explained, $\sigma_{pairwise}^2$, is calculated as $4MAF_1(1-MAF_1)MAF_2(1-MAF_2)\beta_{pairwise}^2$, where the SNP pairs are independent ($LD \approx 0$).

(a) rs72613567-rs738409

(b) rs58542926-rs738409

Figure S14: **Pairwise epistasis test.** We analyze the relationship between (a) rs72613567 and rs738409 and (b) rs58542926 and rs738409 on alanine aminotransferase. The error bar represents the 95% confidence interval from 1,000 bootstrap samples.

6. The study does not include a replication dataset. For the validity and generalizability of the findings, it is crucial to replicate the results in an independent cohort.

We thank the reviewer for pushing us to perform replication analysis which we agree is crucial to establish the robustness of our results. In our revised manuscript, we have performed two replication experiments: an internal replication experiment (within the UKBB) and an external replication experiment in which we attempted to replicate our findings in UKBB in the All of Us data (for those phenotypes that had sufficiently large sample sizes for European ancestry individuals, $N > 50K$). In our internal replication experiment, we discovered three trait-locus pairs with genome-wide significant marginal epistasis (ME) signals in about 150K unrelated white British individuals in the UKBB. We find that all signals replicate in a non-overlapping set of about 150K individuals with consistent direction of effects ($p < 0.05/3$) (see figure below). For our external replication, we selected five traits that had evidence of significant ME in UKBB for which we had a sufficiently large sample size of European ancestry individuals in AoU. We found that all five of our ME signals replicated in AoU with consistent direction of effects ($p < 0.05/5$). These replication experiments substantially strengthen our original results indicating that we are identifying individual SNPs that show strong and replicable signals of polygenic epistasis (see figure below).

Figure 4: **Replicability of the findings from FAME.** We report estimates (and 95% CIs) of σ_{gxg}^2 for genome-wide significant ME loci ($p < \frac{5 \times 10^{-8}}{53}$) discovered in a discovery cohort and their estimates in a replication cohort. (Left). We use as discovery cohort about one half of the unrelated white British individuals in UKBB and as replication cohort a non-overlapping set of unrelated white British individuals in UKBB. b) We use as discovery cohort all unrelated white British individuals in UKBB and as replication cohort individuals of European ancestry in the All of Us (AoU) dataset.

We now include these results in Section 2.71 Replication of ME signals, as:

“We assessed the robustness of FAME signals using an internal and an external replication study.

For our internal replication, we split the unrelated white British individuals in UKBB into a discovery and a replication cohort of equal size. We ran our pipeline in the discovery cohort (where we first performed GWAS followed by application of FAME on each of the traits that exhibited significant ME signals) and identified three ME significant trait-locus pairs.

We found that all signals replicated in the non-overlapping replication set with consistent direction of effects ($p < 0.05/3$), Figure 4).

For our external replication, we attempted to validate our ME signals in the All of Us (AoU) dataset. We identified five traits (of the 13 for which we identified at least one significant ME locus in UKBB) for which we had sufficiently large sample size of European ancestry ($N > 50K$) individuals in AoU (Alanine aminotransferase, Blood Mean Platelet Volume, Blood

Monocyte count, Total cholesterol, and triglycerides). We found that all five of our ME signals replicated in AoU with consistent direction of effects ($p < 0.05/5$, Figure 4).”

7. The study is limited to quantitative traits. The paper should discuss the applicability of the findings to binary traits or include an analysis of such traits for a more comprehensive understanding.

While our method is technically applicable to binary traits, a more detailed understanding of its calibration for low-prevalence traits and the impact of ascertainment is needed for us to be able to analyze binary traits. As a result, our current study focuses on quantitative traits. We have emphasized this in the abstract, writing:

“ We present a method to test for ME of a SNP on a quantitative trait that is applicable to biobank-scale data.”

We further comment on this limitation in the Discussion, writing:

“Third, we have only applied FAME to quantitative traits which match the assumptions underpinning our model. Assessing the applicability of this model to binary (disease) traits would require exploring the impact of trait prevalence and ascertainment.”

8. The paper should clarify how SNP p-values, corrected for marginal effects of other SNPs, relate to biological epistasis. For instance, the association with ApoB, Cholesterol, and TG, and the mediation analyses with other genes, should be contextualized in terms of these corrections. This clarification will aid in understanding the biological implications of the statistical findings.

To address this question from a modeling perspective, we jointly model the genome-wide SNPs' marginal effects along with the target SNP's marginal epistatic effect. At a high level, this approach means that the marginal epistatic component will be significant only if the interaction between the target SNP and the other SNPs (i.e., the marginal epistatic interaction effect) explains additional variation beyond what is captured by the marginal effects alone.

To ensure that our estimates are not confounded by LD or marginal effects, we employed several strategies. First, we excluded SNPs within the LD block from the marginal effect estimation to reduce correlation with the marginal signal. Second, we regressed out the main effects of SNPs within the LD block to prevent confounding between the marginal epistatic signal and the marginal component. Through extensive simulations, we validated that this approach successfully controls false positives (Supplementary Figure S2 that we attached above).

To delve deeper into the biological relevance of the ME loci, we conducted the following additional analyses:

1. Functional enrichment analysis:

- a. For the target ME SNPs, we overlapped our set of genome-wide significant ME signals with sets of eQTLs (cis and trans-eQTLs in whole blood identified in Vosa, Claringbould et al. *Nature Genetics* 2021) and pQTLs (cis and trans pQTLs for 57 plasma proteins identified in Yao et al. *Nat. Comm.* 2018 and 4,907 plasma proteins identified in Ferkingstad et al. *Nature Genetics* 2021). Eleven of the 14 ME SNPs were identified as regulating gene expression or protein levels.

rs146534110, which is a ME SNP for Lipoprotein-A, is a cis-pQTL for LPA. rs964184, which is a ME SNP for three traits (Apolipoprotein B, Cholesterol, and Triglycerides), was found to be a cis-eQTL, a trans-eQTL, and a trans-pQTL. We describe these analyses in Section 2.7.4 titled “Functional interpretation of loci with significant ME effects”.

- b. We performed pairwise epistasis analysis of the 16 genome-wide significant ME signals (GxGWAS). For each GxGWAS result, we selected the top 1K SNPs that interact with the target SNP based on the interactive p-value (excluding SNPs from the LD block of the target SNP and the MHC region). We combined these sets of SNPs across all traits and used RegulomeDB (Boyle et al. *Genome Research* 2012) to test if the interacting SNPs were enriched for functional annotations at the SNP sites (measured as the fraction of sites with rank score $\geq 1f$) relative to a background of GWAS variants pooled across all outcomes (repeated 10,000 times). We identified significant functional enrichment with p-value=0.033. (We describe these analyses in Section 2.7.4 with more details).
 - c. To test whether the interactive SNPs are enriched for binding of specific transcription factors (TFs), we applied HOMER (Heinz et al. *Molecular cell* 2010) and analyzed a ± 100 bp window around the top 1,000 interactive SNPs for TF motif enrichments. We observed significant enrichment for multiple known (FDR<0.05) and de novo ($p < 1 \times 10^{-12}$) TF binding sites. Overall, these analyses revealed enrichments especially for TFs with helix-loop-helix domains as well as for GATA and nuclear hormone receptor TFs with zinc finger domains, which are known to be involved in the regulation of metabolic traits (Wang and Baker. *Developmental cell* 2015; Tong et al. *Science* 2000; Wei et al. *Cellular and molecular life sciences*, 2013).
2. Analyses of individual ME loci: We have now analyzed individual ME loci underlying Lipoprotein A, Lipids, Alanine aminotransferase, and Testosterone in the Section titled “Interpretation of individual ME loci”. As a prominent example, we have detailed the example of the ME SNP associated with alanine aminotransferase in our response above.

9. The term "spearman correlation" is misspelled as "spearsman" in Figure S6. This should be corrected for accuracy.

Thank you, we have fixed this typo.

Reviewer #3:

Remarks to the Author:

A biobank-scale test of marginal epistasis reveals genome-wide signals of polygenic epistasis

Fu et al.

The study introduces a method to estimate SNP-by-genetic background interactions, and investigates the contribution of epistasis to complex human traits. Existing methods aimed at identifying interactions among genetic variants face challenges such as low power and computational issues due to the testing of numerous hypotheses. Additionally, spurious epistasis signals can arise from untagged causal variants, posing a significant concern. The proposed approach appears to effectively address these limitations. Applied to biobank-scale data, the method's calibrated performance is confirmed through simulations. In the examination of 15,601 trait-loci associations in the UK Biobank, the study successfully identifies 16 trait-loci pairs across 12 traits, uncovering robust evidence of marginal epistasis (ME). These results, the first of their kind, underscore the substantial contribution of ME to trait variance, surpassing the impact of Genome-Wide Association Study (GWAS) effects.

While this study is both important and intriguing, there are several comments that should be clearly addressed.

We thank the reviewer for their positive comments.

<Comments>

The accuracy of estimating variance components using the Method of Moments (MoM) can be influenced by several factors, and the use of random variables (B) and simplifications may impact the precision of the estimates. There are several comments on this approach:

1. The choice of these random variables and their characteristics (e.g., distribution, independence) can affect the accuracy and variability of the estimates. Please clearly state the assumptions made for these random variables in the approach. Additionally, elaborate more on this approach in the Results section to help readers understand it better.

We have now clarified our assumptions. For its validity, the randomized MoM estimator only requires that the random vectors are drawn from a distribution with mean zero and identity covariance. In practice, we draw each entry of the vector from an independent standard normal distribution, which is a common technique for constructing stochastic estimators [Hutchinson 1989]. We have clarified this in the section 4.2, Efficient computation of variance components, as:

“We note that $\hat{T}_{i,j}$ is an unbiased estimator of $T_{i,j}$ provided v_b is a random vector with mean zero and covariance I_N . The distribution of v_b can impact the variance of the estimator (Avron and Toledo. J. ACM 2011). In practice, we draw each entry of v_b independently from a standard normal distribution. The variance of $\hat{T}_{i,j}$ decreases with the increasing number of

random vectors (B). Empirically, we observed that utilizing 100 random vectors yields sufficiently accurate approximations for our estimators (Figure S9).”

and also in the Results section as:

“Specifically, FAME utilizes a randomized Method-of-Moments (MoM) estimator that reduces the size of the input genotype and interaction matrices by multiplying each of these matrices with a pre-specified number (B) of random vectors (where each entry of the random vector is drawn independently from a standard normal distribution). This approach is used to approximate key computations in the MoM estimator. The variance of the estimator depends on the number of random vectors B . We show that, even with small values of $B \sim 100$, this approach results in accurate estimates of the variance components resulting in a highly scalable method.”

2. Ad-hoc simplifications in the estimation procedure may reduce the complexity of calculations. However, these simplifications should be carefully justified to ensure they do not introduce bias or compromise the validity of the estimation. It is crucial to validate MoM estimates against known values or alternative methods where possible (e.g., EM, NR, Fisher, or AI algorithms). Using various scenarios and considering assumptions in the MoM approach, the authors should report situations in which MoM estimates deviate from those obtained through exact methods (EM, NR, etc.) so that they can advise readers on when to use MoM and when not to.

The reviewer raises an important point that we would like to elaborate on multiple aspects of: randomized versus exact estimation and MoM versus likelihood-based methods.

With respect to the utility of randomized approaches, we’d like to clarify that both exact and randomized implementations of both MoM and the likelihood-based estimators (typically obtained using the EM, NR, etc. algorithms) are available. We have previously demonstrated that, in the additive context, additional uncertainty introduced by randomization is orders of magnitude smaller than estimator standard errors, both in the MoM (Pazokitoroudi et al. *Nat. Comm.* 2020; AJHG. 2024) and REML NR case (Boarder & Becker. *BMC Bioinformatics.* 2019).

In this manuscript, we have compared MoM estimates to known estimates in simulations and shown that the estimates are calibrated (so that the false positive rate is controlled). In our revised manuscript, we have also explored a number of model violations to show that our method is well-calibrated under these settings, specifically, we performed simulations under this setting of large additive main effects (in addition to several other types of model misspecification, including non-linear covariates that impact the phenotype, GxE with a hidden E variable, heavy-tailed environmental noise, and heteroskedastic environmental noise that we describe in the Section “Robustness to model misspecification” in the main text). We simulated large additive genetic effects and further considered large additive effects that are imperfectly tagged. Finally, we also considered a setting where the dependence of additive effect sizes on MAF and LD is different (LDAK model) than the one assumed by FAME. We find that FAME remains calibrated in each of these settings. We reproduce the QQ plots below and show additional metrics relevant to type-I error rate and λ_{gc} in the Figure S4 and Table S1 below. More details on these simulations are included in Supplementary Text S1.

Figure S4: **Robustness to mis-specification.** We explored the impact of three types of model mis-specifications (Large additive effects, Imperfectly-tagged large additive effects, and Imperfectly-tagged LDAK model) on the calibration of FAME.

α	0.5	0.1	0.05	0.01	0.005	λ_{gc}	CI(λ_{gc})
Polynomial covariate relationship	1.05	1.02	0.88	1.10	0.55	1.11	(0.874, 1.456)
G-E with unobserved E	0.97	0.45	0.69	0.69	0.69	0.95	(0.669, 1.287)
Heavy-tailed noise	0.93	0.71	0.68	0.62	0.93	0.82	(0.607, 1.002)
Heteroskedastic noise	0.98	1.08	1.29	0.54	0.54	0.96	(0.740, 1.232)
Large additive effects	1.02	0.89	0.77	0.77	0.77	1.03	(0.855, 1.194)
Imperfectly-tagged large additive effects	0.97	0.82	0.66	0.73	0.73	0.92	(0.766, 1.144)
Imperfectly-tagged LDAK model	1.02	0.79	0.98	0.75	0.0	1.05	(0.881, 1.439)

Table S1: **Robustness of FAME under model misspecification:** We test the robustness of FAME under seven types of model misspecification. We report the ratio of observed to the expected type-I error rate at different p-value thresholds ($\alpha \in \{0.5, 0.1, 0.05, 0.01, 0.005\}$) and the genomic inflation factor (λ_{gc}). Across all types of model misspecification, the genomic inflation factor is close to 1 (the 95% CI of the inflation factor overlaps 1 across each of the settings). Similarly, the ratio of observed type-I error rate is close to the expected type-I error rate (none of the observed type-I error rate is significantly larger than the expectation across all thresholds and model misspecifications ($p < 0.05$)).

These results point to an important advantage of our approach. Such a detailed exploration of model robustness on realistic-sized datasets is possible because of the efficiency of our method. Thus, our method also enables exploration of models and parameter spaces at a scale that was not previously possible.

With respect to the distinction between MoM and likelihood-based methods, we now emphasize the advantages and limitations of our (randomized) MoM approach compared to the most common MLE or REML alternative that can be estimated using the EM, NR, Fisher or AI algorithms. When the effect sizes and the environmental noise are all normally distributed

(leading to a tractable likelihood), REML is more statistically efficient (having lower standard errors) than MoM. On the other hand, MoM is unbiased under arbitrary distributions on effect sizes and environmental noise so that MoM is more robust to model violation than REML. Most important to our application, MoM is amenable to randomization. The computational efficiency of our randomized MoM enables its application to datasets where, as far as we know, likelihood-based approaches are not feasible. Of course, the question of which estimator to use depends on considerations of assumptions about the model (e.g. which assumptions are likely to be correct) and on dataset size. Randomized implementations of likelihood-based epistasis models remain an interesting topic for future research. We have now added these points to the Discussion Section.

3. When assessing the performance of FAME, showing just a QQ plot may not be sufficient to demonstrate if the type I error rate is genuinely controlled under the null. Please report the genomic inflation factor (aka lambda) and explicitly state the type 1 error rate for all simulation scenarios under the null. Consider including other relevant metrics as well.

We thank the reviewer for this suggestion. We now report λ_{gc} and type-I error rates for all simulations. We also have substantially expanded our simulations to consider model misspecification scenarios that the reviewer points to below (with more details in the responses below). We reproduce the table summarizing our results. In this table, we report the calibration of p -values testing the hypothesis that the epistatic variance component is zero. We report the ratio of observed to the expected type-I error rate at different p -value thresholds ($\alpha \in \{0.5, 0.1, 0.05, 0.01, 0.005\}$) and the genomic inflation factor (λ_{gc}) under the null (both when the model is correctly specified and when the model is mis-specified where the model misspecifications aggregate all the scenarios that we detail below). Across each of these settings, the genomic inflation factor is close to 1 (the 95 % CI of the inflation factor overlaps 1 across each of the settings). Similarly, the ratio of observed type-I error rate is close to the expected type-I error rate (none of the observed type-I error rate is significantly larger than the expectation across all thresholds ($p < 0.05$)).

α	0.5	0.1	0.05	0.01	0.005	0.001	λ_{gc}	CI(λ_{gc})
Correctly specified model	0.94	0.95	0.86	0.60	0.67	0.67	0.88	(0.783, 0.984)
Mis-specified models	0.99	0.83	0.81	0.74	0.68	1.13	0.96	(0.890, 1.043)

Table S2: **Calibration of tests of the hypothesis that $\sigma_{g \times g}^2 = 0$.** We test the calibration of p -values of tests of the hypothesis that $\sigma_{g \times g}^2 = 0$. We report the ratio of observed to the expected type-I error rate at different p -value thresholds ($\alpha \in \{0.5, 0.1, 0.05, 0.01, 0.005\}$) and the genomic inflation factor (λ_{gc}). In the first row, we reported these statistics under a correctly specified null model (causal ratio=0.1, $h^2 = 0.05$ with no ME); in the second row, we report the combined statistics across all settings where the model is mis-specified. Across all types of model misspecification, the genomic inflation factor is close to 1 (the 95% CI of the inflation factor overlaps 1 across each of the settings) while none of the observed type-I error rates is significantly larger than the expectation across all thresholds ($p < 0.05$).

We report these metrics in detail for different model misspecification scenarios in Table S1, which we reproduce in the above response. Across each of these misspecified models, our results suggest that the genomic inflation factor is close to one (i.e., its 95% CI overlaps one)

and none of the observed type-I error rates is significantly larger than the expectation for each threshold.

α	0.5	0.1	0.05	0.01	0.005	λ_{gc}	CI(λ_{gc})
Polynomial covariate relationship	1.05	1.02	0.88	1.10	0.55	1.11	(0.874, 1.456)
G-E with unobserved E	0.97	0.45	0.69	0.69	0.69	0.95	(0.669, 1.287)
Heavy-tailed noise	0.93	0.71	0.68	0.62	0.93	0.82	(0.607, 1.002)
Heteroskedastic noise	0.98	1.08	1.29	0.54	0.54	0.96	(0.740, 1.232)
Large additive effects	1.02	0.89	0.77	0.77	0.77	1.03	(0.855, 1.194)
Imperfectly-tagged large additive effects	0.97	0.82	0.66	0.73	0.73	0.92	(0.766, 1.144)
Imperfectly-tagged LDAK model	1.02	0.79	0.98	0.75	0.0	1.05	(0.881, 1.439)

Table S1: **Robustness of FAME under model misspecification:** We test the robustness of FAME under seven types of model misspecification. We report the ratio of observed to the expected type-I error rate at different p-value thresholds ($\alpha \in \{0.5, 0.1, 0.05, 0.01, 0.005\}$) and the genomic inflation factor (λ_{gc}). Across all types of model misspecification, the genomic inflation factor is close to 1 (the 95% CI of the inflation factor overlaps 1 across each of the settings). Similarly, the ratio of observed type-I error rate is close to the expected type-I error rate (none of the observed type-I error rate is significantly larger than the expectation across all thresholds and model misspecifications ($p < 0.05$)).

The simulation study has some limitations. Please consider the following:

4. The authors should introduce a random variable that interacts with genome-wide SNPs but assume the random variable is unknown in the analysis. In this situation, FAME should be applied in the absence of epistasis, and the authors should assess if the type 1 error rate is controlled. The interaction due to the unknown random variable can be significant, so it should be tested in the range of 0.1 – 0.5 in terms of the phenotypic variance explained by the interaction due to unknown random variables.

We now address this setting in the scenario titled “G-E with unobserved E” (see Table above). The simulation scenario is detailed in Supplementary Information S1 (excerpted below). As the reviewer suggests, the G-E explains substantial phenotypic variance (0.33).

S1.2 G-E with unobserved E

Unknown interactions can affect the model calibration. Such variables can be unobserved environmental variables, missing-tagged causal SNPs, or aggregated PGS.

The generative model for this setting is as follows:

$$\begin{aligned} \mathbf{y} &= \mathbf{X}\boldsymbol{\beta} + \sum_{j \in S} \omega_j \mathbf{X}_{:,j} \odot \mathbf{h} + \boldsymbol{\epsilon} \\ \boldsymbol{\omega} &\sim \mathcal{N}(\mathbf{0}, \frac{\sigma_{hid}^2}{|S|} \mathbf{I}) \\ \boldsymbol{\epsilon} &\sim \mathcal{N}(\mathbf{0}, \sigma_{\epsilon}^2 \mathbf{I}) \end{aligned}$$

Here \mathbf{h} is an unobserved environmental variable (E) drawn from a standard Gaussian distribution and σ_{hid}^2 denotes the variance explained by the hidden G-E. S represents the subset of SNPs selected to interact with

the hidden E. In this experiment, we randomly select 10% of the observed SNPs to interact with the hidden E. We further simulated under two parameter settings $(\sigma_g^2, \sigma_{hid}^2, \sigma_{\epsilon}^2) = (0.1, 0.5, 0.9)$ and $(0.05, 0.5, 0.95)$.

We introduce this configuration in Section 2.3, writing:

We explored the impact of various types of model misspecification on our results. We considered, in turn, scenarios in which ... 2) the genetic architecture includes gene-environment interactions with a hidden environmental variable, ...

And we concluded this finding at the end of the same section, writing:

We found that FAME remains calibrated across each of these settings (Supplementary Figures S3, S4, S5, S6 and Supplementary Tables S1, S2).

5. The authors should consider phenotypic or residual heteroscedasticity, such as heterogeneous residual variances across age, sex, or any other random variables that can be known or unknown. In this situation, FAME should be applied to assess the type 1 error rate as well as the unbiasedness of the estimates.

We thank the reviewer for this suggestion. We now address this in the scenario “Heteroskedastic noise”, detailed in Supplementary Text Section S1.4 (reproduced below).

S1.4 Heteroskedastic noise

We assess the impact of heteroskedastic noise using the following generative model:

$$\begin{aligned} \mathbf{y} &= \mathbf{X}\boldsymbol{\beta} + \mathbf{z} \odot \boldsymbol{\delta} + \boldsymbol{\epsilon} \\ \boldsymbol{\beta} &\sim \mathcal{N}\left(\mathbf{0}, \frac{\sigma_g^2}{M}\mathbf{I}\right) \\ \boldsymbol{\delta} &\sim \mathcal{N}\left(\mathbf{0}, \sigma_{het}^2\mathbf{I}\right) \\ \boldsymbol{\epsilon} &\sim \mathcal{N}\left(\mathbf{0}, \sigma_\epsilon^2\mathbf{I}\right) \end{aligned}$$

Here \mathbf{z} is a binary indicator variable drawn from $Bern(0.5)$, denoting which samples to assign to the group with larger variance. In this experiment, we simulated phenotypes under two parameter settings $(\sigma_g^2, \sigma_{het}^2, \sigma_\epsilon^2) = (0.1, 0.5, 0.9)$ and $(0.05, 0.5, 0.95)$.

Figure S6 (scenario S4) (reproduced below) demonstrates that FAME appears to be unbiased in these settings (estimate of bias = $9.66e-6$; no evidence for significant bias at $p=0.05$).

Figure S6: **Estimator bias under model mis-specification.** We explore the bias of FAME's under the the seven types of model mis-specifications. S1: Polynomial covariate relationship, S2: G-E with unobserved E, S3: Heavy-tailed noise, S4: Heteroskedastic noise, S5: Large additive effects, S6: Imperfectly-tagged large additive effects, S7: Imperfectly-tagged LDAK model. Across all the settings, FAME is relatively unbiased (magnitude of bias $< 1 \times 10^{-4}$).

6. The authors should also introduce large quadratic or polynomial effects of covariates that are not necessarily known. In this situation, FAME should be applied to assess the type 1 error rate as well as the unbiasedness of the estimates.

We have addressed this in the scenario “Polynomial covariate relationship”, detailed in Supplementary Section S1.1 (reproduced below). In brief, we simulated a quadratic effect between age and sex, but did not explicitly model such effect when fitting FAME.

S1.1 Polynomial covariate relationship

This experiment aims to simulate a scenario in which the impact of covariates \mathbf{C} on the phenotype \mathbf{y} is described by a nonlinear (quadratic) function. However, this polynomial relationship is not known. As a result, when we apply FAME, we only include the original untransformed covariates.

The generative model for this setting is as follows:

$$\begin{aligned} \mathbf{y} &= \mathbf{X}\boldsymbol{\beta} + \Phi(\mathbf{C})\boldsymbol{\gamma} + \boldsymbol{\epsilon} \\ \boldsymbol{\epsilon} &\sim \mathcal{N}(\mathbf{0}, \sigma_e^2 \mathbf{I}_N) \\ \boldsymbol{\beta} &\sim \mathcal{N}(\mathbf{0}, \frac{\sigma_g^2}{M} \mathbf{I}_M) \\ \boldsymbol{\gamma} &\sim \mathcal{N}(\mathbf{0}, \frac{\sigma_{cxc}^2}{D} \mathbf{I}_D) \end{aligned}$$

Here \mathbf{X} is the genotype matrix, Φ represents the second-order polynomial transformation (of dimension D), and \mathbf{C} represents the set of covariates (we use age and sex). σ_{cxc}^2 represents the variance explained by the transformed covariates, σ_g^2 represents the variance explained by additive effects, and σ_e^2 represents the noise variance. In this experiment, we simulated phenotypes under two parameter settings $(\sigma_g^2, \sigma_{cxc}^2, \sigma_e^2) = (0.1, 0.25, 0.9)$ and $(0.05, 0.25, 0.95)$.

We also find that FAME appears to be relatively unbiased (estimate of bias = -2.02e-5; no evidence for significant bias at p=0.05). The estimation distribution can be seen from the above reproduced Figure S6 (scenario S1).

Additionally, the authors should consider more comprehensive simulation scenarios to ensure the robustness of their findings and mitigate the possibility of spurious results.

We have now added additional scenarios that include “Heavy-tailed noise”, “Large additive effects”, “Imperfectly-tagged large additive effects”, and “Imperfectly-tagged LDAK model” (Supplementary Section S1 with metrics summarized in the Table above).

Additional comments are as follows:

7. What is the usual range of the magnitude of GxG epistasis? A magnitude of 0.1 for a single genetic variant seems very large. Some references supporting this would be beneficial.

We note that the the largest range of $h^2_{g \times g}$ that we observe is ~0.01. This magnitude (Lipoprotein-A) is for a SNP within the LPA locus which has been shown to explain about 90% of trait variance (Boerwinkle et al. JCI 1992) so that a GxG variance of 0.01 at this locus would not be implausible. We also note that it is likely that we are powered to detect larger effects and that our estimates could be inflated due to Winner’s curse. We state this in the Discussion in our original submission and the results, writing:

“Fifth, our estimates of ME effects are likely to be biased upwards due to winner’s curse [Xiao and Boehnke, 2009], although our replication experiment (Figure 4) suggests that the bias is modest.” (Discussion section)

And *“We observe the largest magnitude of $h^2_{g \times g}$ and the largest ratio of $h^2_{g \times g}$ to h^2_{gwas} at SNP rs628031 for serum lipoprotein A levels (lipoA). This variant lies in a locus (near the LPA gene) that has been shown to explain as much as 90% of trait variance [Boerwinkle et al. 1992].*

rs628031 is a non-synonymous polymorphism in the protein product of the OCT1 gene (also known as SLC22A1). OCT1 mediates the uptake and efflux of cationic metabolites in liver. Genetic variation in OCT1 has been shown to modulate the response to metformin and other drugs [Shu et al. 2007].” (Section 2.7.4, Interpreting loci with significant ME effects)

To further establish the robustness of our findings, we have undertaken a new set of replication analysis. In our revised manuscript, we have performed two replication experiments: an internal replication experiment (within the UKBB) and an external replication experiment in which we attempted to replicate our findings in UKBB in the All of Us data (for those phenotypes that had sufficiently large sample sizes for European ancestry individuals, $N > 50 K$). In our internal replication experiment, we discovered three trait-locus pairs with genome-wide significant marginal epistasis (ME) signals in about 150K unrelated white British individuals in the UKBB. We find that all signals replicate in a non-overlapping set of about 150K individuals with consistent direction of effects ($p < 0.05/3$) (see figure below). For our external replication, we selected five traits that had evidence of significant ME in UKBB for which we had a sufficiently large sample size of European ancestry individuals in AoU. We found that all five of our ME signals replicated in AoU with consistent direction of effects ($p < 0.05/5$). These replication experiments substantially strengthen our original results indicating that we are identifying individual SNPs that show strong and replicable signals of polygenic epistasis (see figure below).

Figure 4: **Replicability of the findings from FAME.** We report estimates (and 95% CIs) of $\sigma_{g \times g}^2$ for genome-wide significant ME loci ($p < \frac{5 \times 10^{-8}}{53}$) discovered in a discovery cohort and their estimates in a replication cohort. (Left). We use as discovery cohort about one half of the unrelated white British individuals in UKBB and as replication cohort a non-overlapping set of unrelated white British individuals in UKBB. b) We use as discovery cohort all unrelated white British individuals in UKBB and as replication cohort individuals of European ancestry in the All of Us (AoU) dataset.

8. The authors should discuss other similar approaches. For instance, a previous study employed the Hadamard product between two genomic relationship matrices to identify interactions between two sets of SNPs:

"Efficient Algorithms for Calculating Epistatic Genomic Relationship Matrices. *Genetics*, Volume 216, Issue 3, 1 November 2020, Pages 651–669, <https://doi.org/10.1534/genetics.120.303459>."

Similarly, other studies have used Cholesky decomposition of two genomic relationship matrices (e.g., regulatory vs. DHS genomic regions) to investigate if their effects are positively or negatively correlated, indicating some interaction between two genomic regions:

"CORE GREML for estimating covariance between random effects in linear mixed models for complex trait analyses. *Nature Communications* volume 11, Article number: 4208 (2020)."

All these studies employed a linear mixed model, similar to the context of this study.

We thank the reviewer for pointing us to these works. The first work (Jiang and Reif, *Genetics*, 2020 as well as the work by Hivert et al. *AJHG* 2021) focuses on estimating the proportion of variance of a quantitative trait explained by genome-wide epistasis. We now clarify that this is distinct from the analysis of marginal epistasis that is our focus. Further, the Hadamard product approach still requires computing relatedness matrices and therefore has a runtime complexity that scales as $O((\text{sample size})^2 \times \text{number of variants})$ (with additional computational complexity depending on whether REML estimates are computed), limiting biobank-scale application; for example, Hivert et al. meta-analyzed multiple estimates across subsamples, substantially reducing power. On the other hand, Jiang and Reif consider models of epistasis that go beyond pairwise epistasis that is our focus here and show how the Hadamard products can be computed in these settings. An interesting direction for future work is whether the ideas that we propose here can be used to test and quantify genome-wide epistasis as well as higher-order epistasis.

As the reviewer points out, the second work extends the additive linear mixed model to a setting where the effect sizes are correlated (e.g. attempting to model the covariance of the effect size of a variant in a protein-coding gene and its regulatory regions). We now clarify the relationship between these approaches and our method in the section 2.1 (Methods overview), writing:

"Fitting this model to Biobank-scale data, containing hundreds of thousands of individuals and millions of SNPs, is computationally impractical. For example, one class of works that aim to efficiently compute genome-wide epistatic genomic relationship matrices (Jiang and Reif, *Genetics* 2020) and (Hivert et al. *AJHG* 2021) still scale quadratically with sample size (while not being directly relevant to estimating ME). The CORE GREML approach (Zhou et al. *Nat. Comm.* 2020) explicitly estimates covariance between random effects but does not directly model epistasis. Finally, a previous work that tests for ME does not scale to large samples (Crawford et al. *PLoS Genetics* 2017). FAME expands on our recent work (Wu and Sankararaman, *Bioinformatics* 2018) and (Pazokitoroudi et al. *Nat. Comm.* 2020) to be able to test and estimate ME on Biobank-scale data."

9. In line 136, it is stated, “We observed that FAME remains calibrated indicating its robustness to imperfect tagging (Supplementary Figure S3).” As mentioned in #3 above, please explicitly report the type 1 error rate, genomic inflation factor, and the distribution of estimated GxG. Additionally, as per comment #4, assign large interaction effects on untagged SNP(s) to observe how the type 1 error rate is affected.

We have now updated and expanded these simulations to incorporate both imperfect tagging and large marginal effects into the scenario titled “Imperfectly-tagged large marginal effects” (in the Table above and detailed in Supplementary Information S1).

10. It is not clear how the authors simulated the data under imperfect tagging scenarios. The authors should also consider untagged haplotype scenarios, as pointed out in reference #45. We note that our simulations are a generalization of the untagged haplotype scenario in Wood et al. (ref #45). Wood et al. showed that the findings of epistatic effects of cis-cis SNP pairs on gene expression probes (pairs of SNPs that lie close to each other on the same chromosome) were removed when controlling for a third variant that was strongly associated with the probe. They showed that it was possible to have settings where the cis-cis SNP pair has no LD although each of these SNPs were in moderate LD with the third SNP (so that LD pruning would not eliminate such effects).

For the imperfect tagging scenarios, we simulated phenotypes using imputed SNPs while, during the estimation stage, we assumed only SNPs on the UK Biobank array were observed (both the target SNP and its interacting partners). Further, we also considered scenarios with large additive effects of the causal variants as well as a scenario where the causal variants were drawn from the LDAK model (Speed et al. *AJHG* 2012) which posits a different relationship between effect size, MAF, and LD than is assumed by our model (which is based on the GCTA model; Yang et al. *Nature Genetics* 2010).

We have included these imperfect tagging scenarios in Section 2.3, Robustness to model misspecification. The detailed calibration results are provided in Table S1 in the above response. More details on the simulation design can be found at Supplementary text S1.6 and S1.7, reproduced below.

S1.6 Imperfectly-tagged large additive effects

This experiment is similar to the Large additive effects setting (Section S1.5) except that we relaxed the assumption that all the extremely large effect variants were observed, acknowledging the presence of imperfectly tagged SNPs. Specifically, we simulated phenotypes using the imputed SNPs (*UKBB-small-imputed*) while during the estimation stage, we assumed only the SNPs in *UKBB-small* array data were observed.

S1.7 Imperfectly-tagged LDAK model

This experiment aims to study the calibration of FAME when the true genetic architecture has a different relationship between effect size, MAF, and LD (LDAK model [80]) than the one assumed in FAME. In addition, we assume that not all causal SNPs are observed.

$$\begin{aligned}\mathbf{y} &= \mathbf{X}\boldsymbol{\beta} + \boldsymbol{\epsilon} \\ \boldsymbol{\beta} &\sim \mathcal{N}(\mathbf{0}, \text{diag}(\sigma_{g1}^2, \dots, \sigma_{gM}^2)) \\ \boldsymbol{\epsilon} &\sim \mathcal{N}(\mathbf{0}, \sigma_{\epsilon}^2 \mathbf{I})\end{aligned}$$

We assumed the LDAK genetic architecture. Specifically, $\sigma_{gm}^2 = Sc_m w^b [f_m(1 - f_m)]^a$, where S is the normalization constant such that $\sum_{m=1}^M \sigma_{gm}^2 = \sigma_g^2$; c_m is an indicator variable indicating whether SNP m is causal; f_m is the MAF of SNP m ; w_m is the LD score of SNP m , and $a = 0.75$, $b = 1$ for the LDAK model. Further, we simulated the phenotype using imputed genotypes (*UKBB-small-imputed*). However, for estimation, we used the array version of *UKBB-small*, a subset of the imputed version, to mimic scenarios with missing-causal SNPs. We set $(\sigma_g^2, \sigma_{\epsilon}^2) = (0.05, 0.95)$ and $(0.1, 0.9)$, the causal SNP ratio was set to 0.1.

11. Regarding the scale of y in line 211, it is mentioned, “We noticed that by changing the scale of the target trait, the p-values from FAME were significantly inflated (Supplementary Figure S9), thus suggesting that the ME signals discovered by FAME are unlikely to arise due to the scale on which traits are measured.”. However, this does not necessarily rule out the possibility of inflation due to scale, as it depends on the level of the scale. Could the authors explore other transformations (e.g., $y^{\wedge}1.1$, ...) to observe how the genomic inflation factor changes? This experimentation can also be conducted through simulations to quantify alterations in the type 1 error rate. While this is related to question #5, it should be separately and explicitly tested.

We have now performed this simulation (transforming the trait by a varying exponent) and do find systematic inflation as the exponent becomes sufficiently large. However, we note that we always transform the trait by applying a rank-based inverse normal transformation (IVRT) in simulations and real data analyses. The trait values (after IVRT) remain invariant to the scaling by an exponent (or other monotone transformations) so that the IVRT-transformed phenotype remains calibrated for all values of the exponent.

Figure S12: **Scaling effect of simulated phenotype on ME estimates.** We explored the impact of phenotype scale on the p-value estimated by FAME in simulations. We simulated phenotypes using genotype data from the *UKBB-small* with additive architectures (10% of the causal SNPs explaining 0.05 heritability). Next, we performed inverse rank normal transformation on the phenotype and transformed it as $sign(y) * |y|^s$, where s is a scaling factor. Finally, we randomly selected 200 GWAS hits after LD pruning as the ME candidate and applied FAME to estimate ME at these SNPs. The figure shows the inflation of FAME with varying scaling factors ($s = 1$ is the original unscaled phenotype). We also show the results applied to the phenotype after inverse rank normalization (IVRT); this transformation is invariant to scaling).

We discuss this in Section 2.6.2, writing:

“Impact of phenotype scale: *The scale on which the phenotype is measured can affect approaches to test and interpret epistasis (Sverdlov and Thompson, bioRxiv 2018). To explore the impact of scale, we performed null simulations where the phenotype has an additive genetic architecture but no genetic interactions. Next, we considered transformations of the phenotype: $sign(y) * |y|^s$, where s is a scaling factor. We randomly selected 200 LD-pruned GWAS hits and applied FAME to estimate ME at these SNPs. The figure shows the inflation of the p-values estimated by FAME with increasing magnitude of the scaling factor ($s=1$ is the original unscaled phenotype; Supplementary Figure S12). In all our analyses, traits are inverse rank normalized so that the results of FAME are invariant to monotone transformations (such as the scaling transformation considered here) and the p-values remain calibrated, as we confirm in simulations (Supplementary Figure S12).”*

Reviewer #1 (Remarks to the Author):

I appreciate the authors' detailed responses to my comments. The addition of independent replication data and other changes to the manuscript have strengthened it substantially.

We thank the reviewer for their positive comments.

My remaining comments are relatively minor:

Figure 3. I don't understand what the $h^2_{gwas,t}$ quantity represents or how it is calculated. Is this just the additive variance explained by the single genome-wide significant SNP? OK, reading further in the manuscript it seems like this is the single locus variance. Perhaps this can be made clear to the reader in the figure legend so the reader doesn't have to go searching for it?

We apologize for not making the notation and the figure clear. The computation of $h^2_{gwas,t}$ and its SE was previously described in Supplementary Information S3. We have now clarified the meaning of $h^2_{gwas,t}$ in the revised Figure 3 and its caption (reproduced below).

(a) Manhattan plot of FAME p-value distribution

(b) Localization of ME signals

(c) The fraction of phenotypic variance explained by marginal epistatic effects ($h_{gxg,t}^2$) vs that explained by additive effects tagged by the genome-wide significant SNP t ($h_{gwas,t}^2$) for each of the 16 ME significant trait-locus pairs

Figure 3: ME signals in the UKBB. (a) Manhattan plot of the ME loci across 53 complex traits in UKBB. Colored shapes denote trait-locus pairs that are significant at $p < \frac{5 \times 10^{-8}}{53}$; shapes with colored triangles were the loci that are statistically significant in our initial analysis and after we regressed out all SNPs within the LD block as fixed effects. (b) Localization of ME signals. For each of 16 trait-locus pairs, we tested whether the ME signals remained significant when testing against all SNPs on the same chromosome as the target SNP (after removing SNPs in the same LD block as the target SNP), which we term *local*, and against all SNPs on chromosomes different from the chromosome containing the target SNP, which we term *distal*. We then compared the overlap between the *local* and *distal* significant signals ($p < \frac{5 \times 10^{-8}}{53}$). (c) We compared the fraction of phenotypic variance explained by marginal epistatic effects ($h_{gxg,t}^2$) to the fraction of phenotypic variance explained by additive effects tagged by the single genome-wide significant SNP t (denoted as the $h_{gwas,t}^2$) for trait-locus pairs that show significant ME. Vertical (horizontal) bars denote the standard error of $h_{gxg,t}^2$ ($h_{gwas,t}^2$). We describe the estimation of $h_{gwas,t}^2$ and its SE in Supplementary Information S3. The Pearson correlation between estimates of $h_{gxg,t}^2$ and $h_{gwas,t}^2$ is $\rho = 0.02$.

The standard errors for e.g. the top leftmost SNP (cardiovascular) and top rightmost SNP (also cardiovascular) look odd. The SE is very small for $h^2_{gwas,t}$ for the top left SNP in comparison to the same quantity for the top rightmost SNP, yet the SE for $h^2_{gwas,t}$ is negligible for the top-left SNP but huge for the top right SNP. Is this correct? I'm confused as to how this can be the case?!

To better understand this observation, we consider the estimates for $h^2_{gwas,t}$ and its SE that was previously described in Supplementary Information S3. Dropping the hats on estimators for readability, we have $h^2_{gwas,t} = \beta_t^2$ and $SE(h^2_{gwas,t}) = 2|\beta_t| SE(\beta_t)$ where β_t is the GWAS effect size associated with the standardized genotype at SNP t . For standardized phenotypes (as is the default in our analyses), we have $SE(\beta_t) \approx \frac{1}{\sqrt{N}}$

where N is the number of individuals so that $SE(h^2_{gwas,t}) \approx \frac{2}{\sqrt{N}} \sqrt{h^2_{gwas,t}}$.

This relationship explains why the SEs are expected to grow with the point estimate as we see in Figure 3c.

“For example, one class of works...” Rephrase?

We have now rephrased this sentence to read: “For example, previous approaches that aim to efficiently compute genome-wide epistatic genomic relationship matrices [28] still scale quadratically with sample size (while not being directly relevant to estimating ME).”

Is it surprising that there doesn't seem to be a relationship between $h^2_{gwg,t}$ and $h^2_{gwas,t}$? What is $\rho = 0.02$ in this figure? Is this the correlation between the estimates? If so, this should be described in the figure legend.

ρ refers to the Pearson correlation coefficient between the proportion of variance explained by ME and GWAS. We have clarified the Figure 3 caption to read “The Pearson correlation between estimates of $h^2_{gwg,t}$ and $h^2_{gwas,t}$ is $\rho = 0.02$.” (see revised Figure 3 that we replicated above).

Our observation of a low correlation does not allow us to make strong general statements on the general relationship between $h^2_{gwg,t}$ and $h^2_{gwas,t}$ as this observation is based on a limited set of 16 trait-SNP pairs which have been ascertained based on GWAS. Understanding this relationship generally will require us to estimate ME effects genome-wide and compare them to GWAS effects.

From a methodological perspective, the FAME model does not require a large main effect to have power to detect ME. To validate this, we utilized the same simulations used in our power analysis (utilizing the UKBB-small dataset consisting of 32,708 SNPs from chromosomes 12 and 20). In these simulations, additive effects were chosen independently of the target ME SNP. We ran FAME on all SNPs and also performed a marginal association analysis on the target variants. Among the significant ME loci detected by FAME ($p < 5e-08$), 32.5% were also significant in main effect, while 27.6% had main effect p-values > 0.05 . This analysis demonstrates that FAME can detect interactions even in the absence of strong main effects.

Table 2. Gene names in italics please. “Effect allele” not “Effective allele”

We have italicized the gene names both in Table 2 and the main text and made the edit.

What constitutes an “LD block” needs to be defined more precisely.

LD blocks are defined based on prior work [Berisa and Pickrell, *Bioinformatics*, 2016] which we now clarify this in the text (when first discussing LD block) by stating “*We used LD blocks that were identified in the 1000 Genomes European population by Berisa and Pickrell [32]*”.

Imperfectly tagged causal SNPs: “We observed concordance of the estimates on the imputed dataset with those obtained on the array dataset (Supplementary Figure S10) with the two loci for SHBG showing significantly lower ME estimates.” Why does the estimate of variance explained by the epistatic component decrease? This seems counterintuitive, e.g. when additional (imputed) SNPs are included in an additive GRM often the SNP heritability increases since more of the genome is tagged, but the opposite appears to be the case here. Why?

The variance explained by marginal epistasis (ME) from array data may not be directly comparable to imputed data and the apparent underestimation in imputed data can arise through multiple mechanisms. A critical factor influencing this observation is the genome-wide distribution of ME effects, particularly their polygenicity and relationship with MAF and LD. We identify several key reasons that could explain these differences:

First, when ME effects are predominantly represented in array SNPs, additional imputed SNPs may dilute the signal. To see this, we simulated ME effects for a target SNP (rs3181300, MAF~0.5) interacting with SNPs on the genotyping array from chromosomes 12 and 20 (*UKBB-small* dataset that we describe in our manuscript). When estimating these effects using imputed SNPs (*UKBB-small-imputed* dataset that we also describe in our manuscript), we observed consistently lower ME estimates compared to the ground truth (averaged across 50 replicates). This observation contrasts with our previous demonstration of approximately unbiased estimation when using array SNPs (Main Figure 2d).

Second, fitting a single ME component (our approach in this study) might be biased due to the implied assumption about the relationship between effect sizes, MAF, and LD. Such biases have been observed by us and others for the task of estimating additive genetic variance. For example, estimates of additive heritability are lower on imputed vs array SNPs across multiple quantitative traits in UKBB when we fit a single genetic variance component using our method, RHE-mc [Pazokitoroudi et al, *Nature Communications*, 2020] (see Figure below). We hypothesize that ME effects may exhibit strong dependencies with LD and other patterns so that using a single component to model imputed data may not provide the most accurate estimates.

With these caveats, we view the overall concordance of ME estimates across array vs imputed SNPs as evidence of the robustness of our approach. A deeper understanding of the architecture of ME and our ability to estimate ME from different classes of

variants) is of high interest and we look forward to addressing this question with WGS datasets that are now being generated.

“We found that all five of our ME signals replicated in AoU with consistent direction of effects”. I don’t understand how there is a direction of effect in the epistatic part of the analysis? Also, I'd like to see p-values for all of these tests not just $p < \text{some threshold}$. By “consistent direction of effects”, we meant to express that the pattern of strength of the signal of AoU is roughly consistent with that of the UKBB. For example, Alanine-aminotransferase-rs3827385 being the relatively highest ME while Monocyte count - rs79490353 being the lowest. We feel that this wording was confusing and have removed the phrase “with consistent direction of effects”.

We attached the detailed p-value of All of Us ME analysis below which we now also report in Supplementary Table S6 (reproduced below).

Trait	SNP ID	$\sigma_{g_{xg,t}}^2$ $\times 0.001$	$SE(\sigma_{g_{xg,t}}^2)$ $\times 0.001$	$p_{g_{xg,t}}$
Alanine aminotransferase	rs3827385	12.85	3.37	1.37×10^{-4}
Monocyte count	rs79490353	3.49	1.30	7.12×10^{-3}
Mean platelet volume	rs463312	5.67	1.87	2.39×10^{-3}
Cholesterol	rs964184	11.42	3.33	6.05×10^{-4}
Triglycerides	rs964184	10.96	3.34	1.03×10^{-3}

Table S6: **ME signals in the AoU dataset.** We reported the estimated marginal epistasis variance component $\sigma_{g_{xg,t}}^2$, the corresponding standard error $SE(\sigma_{g_{xg,t}}^2)$, and the p-value significance as $p_{g_{xg,t}}$.

“The significant pairwise effects were almost always located on the same chromosome and close to the target SNP (Supplementary Table S7).” I think it is worth reporting these results in more detail.

Are the significant SNPs close by the index locus? This could point to the existence of e.g. long-range haplotype effects that are being picked up as “epistasis”.

We would first like to highlight that, in reporting significant SNPs, we excluded those within the LD block of the target (index) locus. Further, we quantified the distributions of LD (r^2) (box plot colored with blue; with mean: 0.0019; median: 0.0005; min: 0; max: 0.0176) and physical distance (box plot colored with orange; mean: 699.3 kb; median: 755.4 kb; min: 5.7 kb; max: 1359.9 kb) for the significant SNPs with the target variant ($p \leq 5 \times 10^{-8} / 16$) across the four trait-SNP pairs that demonstrated strong local interaction enrichment (Supplementary Figure S14 reproduced below). Overall, the r^2 values indicate that these SNPs are not highly correlated ($r^2 < 0.10$) suggesting that the

marginal epistatic effect we detected is unlikely to be driven by spurious correlation. We have now added these details to the main text under section 'Localizing signals of ME'.

More broadly, we argue that long-range haplotype effects are unlikely to be a major factor influencing our results, based on the following considerations:

1. In one of our simulation settings, the imperfectly-tagged model, we simulated similar conditions and demonstrated that FAME remains well-calibrated under such scenarios (Supplementary Fig. S4).
2. The majority of the detected ME pairs (11/16) do not show any pairwise significance across the entire genome, suggesting that statistical correlation arising from long-range haplotype effects alone are unlikely to drive the overall marginal epistasis significance.

Figure S14: **Statistics for the significant SNP pairs where the interactive SNP is located on the same chromosome as the target SNP.** We report the physical distance (left), and LD (r^2) (right) between the interactive SNPs and the target SNPs for the significant pairwise interactions across four trait-locus pairs with significant ME (LipoA-rs628031, LipoA-rs146534110, SHBG-rs11656323, and SHBG-rs35985803).

“rs72654473, downstream of the APOE gene on chromosome 9, shows a significant genetic interaction effect with rs6935921, near the LPA gene on chromosome 6.” I think it would be worth graphing this interaction in a figure to see what its form is?

We have now plotted this interaction. It can be seen that the rs6935921-T allele attenuates the effect of rs72654473 on LipoA . We now state this in the text and have now added this as Supplementary Figure S14.

“We overlapped the 14 unique SNPs showing evidence of significant ME effects with sets of eQTLs and pQTLs”. I’m not suggesting the following for this manuscript, but the authors should take advantage of the UKBB proteomic data to go looking for epistatic interactions there, particularly the ones identified in this manuscript as further evidence for their existence.

We thank the reviewers for this suggestion. We would like to leave the exploration of the rich proteomic data in UKBB as well as newer functional genomic datasets as a future direction.

Reviewer #2 (Remarks to the Author):

The authors have demonstrated a commendable approach to addressing the previous review comments. I appreciate the thorough revisions, which substantially improve the manuscript's rigor and clarity:

1. Replication Datasets: The addition of both external and internal dataset replication analyses significantly strengthens the manuscript's methodological validation. This approach provides robust evidence for the proposed method's generalizability and reproducibility.
2. Simulation Scenarios: The expanded simulation scenarios offer a more comprehensive examination of the method's performance under various conditions, which enhances the manuscript's methodological credibility.

3. Clarity and Limitations: The authors have successfully addressed previous points of ambiguity and provided a transparent discussion of the method's limitations.

While potential avenues for future research exist — such as exploring long-range linkage disequilibrium, extending the calibration to binary traits, and investigating non-additive genetic encoding — these opportunities do not diminish the significant scientific contribution of the current work. On the contrary, they highlight the innovative nature of the current manuscript and provide a clear roadmap for future methodological developments. The manuscript represents a critical advancement in marginal epistasis analysis for large biobank-scale datasets, offering a high throughput approach that will undoubtedly stimulate further research and methodological innovations in the field.

We thank the reviewer for their support.

Reviewer #2 (Remarks on code availability):

It would be useful to give examples of input and output datasets.

We have now provided example input and output data at <https://github.com/sriramlab/FAME/tree/main/example>

Reviewer #3 (Remarks to the Author):

None.